# A Dynamic Low-Rank Fast Gaussian Transform

## Abstract

The *Fast Gaussian Transform* (FGT) enables subquadratic-time multiplication of an $n \times n$ Gaussian kernel matrix $\mathsf{K}_{i,j} = \exp(-\|x_i - x_j\|_2^2)$ with an arbitrary vector $h \in \mathbb{R}^n$, where $x_1, \ldots, x_n \in \mathbb{R}^d$ are a set of *fixed* source points. This kernel plays a central role in machine learning and random feature maps. Nevertheless, in most modern data analysis applications, datasets are dynamically changing (yet often have low rank), and recomputing the FGT from scratch in (kernel-based) algorithms incurs a major computational overhead ($\gtrsim n$ time for a single source update $\in \mathbb{R}^d$). These applications motivate a *dynamic FGT* algorithm, which maintains a dynamic set of sources under *kernel-density estimation* (KDE) queries in *sublinear time* while retaining Mat-Vec multiplication accuracy and speed.

Assuming the dynamic data-points $x_i$ lie in a (possibly changing) $k$-dimensional subspace ($k \leq d$), our main result is an efficient dynamic FGT algorithm, supporting the following operations in $\log^{O(k)}(n/\varepsilon)$ time: (1) Adding or deleting a source point, and (2) Estimating the "kernel-density" of a query point with respect to sources with $\varepsilon$ additive accuracy. The core of the algorithm is a dynamic data structure for maintaining the *projected* "interaction rank" between source and target boxes, decoupled into finite truncation of Taylor and Hermite expansions.

## 1 Introduction

The fast Multipole method (FMM) was described as one of the top 10 most important algorithms of the 20th century (Dongarra & Sullivan, 2000). It is a numerical technique that was originally developed to speed up calculations of long-range forces for the $n$-body problem in theoretical physics. FMM was first introduced in 1987 by Greengard and Rokhlin (Greengard & Rokhlin, 1987), based on the multipole expansion of the vector Helmholtz equation. By treating the interactions between far-away basis functions using the FMM, the underlying matrix entries $M_{ij} \in \mathbb{R}^{n \times n}$ (encoding the pairwise "interaction" between $x_i, x_j \in \mathbb{R}^d$) need not be explicitly computed nor stored for matrix-vector operations – This technique allows to improve the naïve $O(n^2)$ matrix-vector multiplication time to quasi-linear time $\approx n \cdot \log^{O(d)}(n)$, with negligible (polynomial-small) additive error.

Since the discovery of FMM in the late 80s, it had a profound impact on scientific computing and has been extended and applied in many different fields, including physics, mathematics, numerical analysis and computer science (Greengard & Rokhlin, 1987; Greengard, 1988; Greengard & Rokhlin, 1988; 1989; Greengard, 1990; Greengard & Strain, 1991; Engheta et al., 1992; Greengard, 1994; Greengard & Rokhlin, 1996; Beatson & Greengard, 1997; Darve, 2000; Yang et al., 2003; 2004; Martinsson, 2012; Chandrasekaran et al., 2006). To mention just one important example, we note that FMM plays a key role in efficiently maintaining the SVD of a matrix under low-rank perturbations, based on the Cauchy structure of the perturbed eigenvectors (Gu & Eisenstat, 1994). In the context of machine learning, the FMM technique can be extended to the evaluation of matrix-vector products with certain *Kernel matrices* $\mathsf{K}_{i,j} = f(\|x_i - x_j\|)$, most notably, the *Gaussian Kernel* $\mathsf{K}_{i,j} = \exp(-\|x_i - x_j\|_2^2)$ (Greengard & Strain, 1991). For any query vector $q \in \mathbb{R}^n$, the *fast Gaussian transform* (FGT) algorithm outputs an arbitrarily-small *pointwise additive* approximation to $\mathsf{K} \cdot q$, i.e., a vector $z \in \mathbb{R}^n$ such that $\|\mathsf{K} \cdot q - z\|_\infty \leq \varepsilon$, in merely $n \log^{O(d)}(\|q\|_1/\varepsilon)$ time, which is dramatically faster than naïve matrix-vector multiplication ($n^2$) for constant dimension $d$. Note that the (poly)logarithmic dependence on $1/\varepsilon$ means that FGT can achieve *polynomially-small*

additive error in quasi-linear time, which is as good as exact computation for all practical purposes. The crux of FGT is that the $n \times n$ matrix K can be stored *implicitly*, using a clever spectral-analytic decomposition of the geometrically-decaying pairwise distances ("interaction rank", more on this below).

Kernel matrices play a central role in machine learning (Shawe-Taylor & Cristianini, 2004; Rahimi & Recht, 2008), as they allow to extend convex optimization and learning algorithms to nonlinear feature spaces and even to non-convex problems (Li & Liang, 2018; Jacot et al., 2018; Du et al., 2019; Allen-Zhu et al., 2019a;b; Lee et al., 2020). Accordingly, matrix-vector multiplication with kernel matrices is a basic operation in many ML optimization tasks, such as Kernel PCA and ridge regression (Alaoui & Mahoney, 2015; Avron et al., 2017a;b; Lee et al., 2020), Gaussian-process regression (GPR) (Rasmussen & Nickisch, 2010), Kernel linear system solvers (via Conjugate Gradient (Alman et al., 2020)), and in fast implementation of the dynamic "state-space model" (SSM) for sequence-correlation modeling (which crucially relies on the Multipole method (Gu et al., 2021)), to mention a few. The related data-structure problem of *kernel density estimation* of a point (Charikar & Siminelakis, 2017; Backurs et al., 2018; Charikar & Siminelakis, 2019; Charikar et al., 2020; Zandieh et al., 2023; Alman & Song, 2023) $\mathsf{KDE}(X, y) = \frac{1}{n} \sum_{i=1}^{n} \mathsf{K}(x_i, y)$ has various applications in data analysis and statistics (Fan & Gijbels, 1996; Schölkopf & Smola, 2002; Schubert et al., 2014), and is the main subroutine in the implementation of transfer learning using kernels (see (Charikar & Siminelakis, 2017; Charikar et al., 2020) and references therein, and the Related Work Section 2 below). As such, speeding up matrix-vector multiplication with kernel matrices, such as FGT, is an important question in theory and practice.

One drawback of FMM and FGT techniques, however, is that they are *static* algorithms, i.e., they assume a *fixed* set of $n$ data points $x_i \in \mathbb{R}^d$. By contrast, most aforementioned ML and data analysis applications are *dynamic* by nature and need to process rapidly-evolving datasets to maintain prediction and model accuracy. One example is the renewed interest in *online regression* (Cohen et al., 2015; Jiang et al., 2022), motivated by *continual learning* theory (Parisi et al., 2019). Indeed, it is becoming increasingly clear that many static optimization algorithms do not capture the requirements of real-world applications (Jain et al., 2008; Chen et al., 2020b;a; Song et al., 2021a;b; Xu et al., 2021; Shrivastava et al., 2021). Notice that changing a single source-point $x_i \in \mathbb{R}^d$ generally affects an *entire row* ($n$ distances $\|x_i - x_j\|$) of the matrix K. As such, naively re-computing the static FGT on the modified set of distances, incurs a prohibitive computational overhead ($n \gg d$). This raises the natural question of whether it is possible to achieve *sublinear*-time insertion and deletion of source points, as well as "local" *kernel-density estimation* (KDE) queries (Charikar & Siminelakis, 2017; Yang et al., 2003), while maintaining speed and accuracy of matrix-vector multiplication queries:

> *Is it possible to 'dynamize' the Fast Gaussian Transform, in sublinear time? Can the exponential dependence on $d$ (Greengard & Strain, 1991) be mitigated if the data-points $x_i$ lie in a $k$-dimensional subspace of $\mathbb{R}^d$?*

The last question is motivated by the recent work of (Cherapanamjeri & Nelson, 2022), who observed that kernel-based methods and algorithms typically involve *low-rank* datasets, (where the "intrinsic" dimension is $w \ll d$), in which case one could hope to circumvent the exponential dependence on $d$ in the aforementioned (static) FMM algorithm (Greengard & Strain, 1991; Alman et al., 2020).

## 1.1 MAIN RESULT

Our main result is an affirmative answer to the above question. We design a fully-dynamic FGT data structure, supporting *polylogarithmic*-time updates and "density estimation" queries, while retaining quasi-linear time for arbitrary Mat-Vec queries (K$q$). More formally, for a set of $N$ "source" points $s_1, \dots, s_N$, the $j$-th coordinate $(\mathsf{K}q)_{j \in [N]}$ is $G(s_j) = \sum_{i=1}^{N} q_i \cdot e^{-\|s_j - s_i\|_2^2 / \delta}$, which measures the kernel-density at $s_j$ ("interaction" of $s_j$ with the rest of the sources). More generally, for any "target" point $t \in \mathbb{R}^d$, let $G(t) := \sum_{i=1}^{N} q_i \cdot e^{-\|t - s_i\|_2^2 / \delta}$ denote the *kernel density* of $t$ with respect to the sources, where each source $s_i$ is equipped with a *charge* $q_i$. Our data structure supports fully-dynamic source updates and density-estimation queries in *sublinear* time. Observe that

this immediately implies that entire Mat-Vec queries ($\mathsf{K} \cdot q$) can be computed in quasi-linear time $N^{1+o(1)}$. The following is our main result:

**Theorem 1.1** (Dynamic Low-Rank FGT, Informal version of Theorem F.2). *Let $\mathcal{B}$ denote a $w$-dimensional subspace $\subset \mathbb{R}^d$. Given a set of source points $s$, and charges $q$, there is a (deterministic) data structure that maintains a fully-dynamic set of $N$ source vectors $s_1, \cdots, s_N \in \mathcal{B}$ under the following operations:*

- INSERT/DELETE($s_i \in \mathbb{R}^d, q_i \in \mathbb{R}$) *Insert or Delete a source point $s_i \in \mathbb{R}^d$ along with its "charge" $q_i \in \mathbb{R}$, in $\log^{O(w)}(\|q\|_1/\varepsilon)$ time. The intrinsic subspace $\mathcal{B}$ could change as the source points are updated.*

- DENSITY-ESTIMATION($t \in \mathcal{B}$) *For any point $t \in \mathcal{B} \subset \mathbb{R}^d$, output the kernel density of $t$ with respect to the sources, i.e., output $\widetilde{G}$ such that $G(t) - \varepsilon \leq \widetilde{G} \leq G(t) + \varepsilon$ in $\log^{O(w)}(\|q\|_1/\varepsilon)$ time.*

We note that when $w = d$, the costs of our dynamic algorithm match the static FGT algorithm. As one might expect, our data structure applies to a more general subclass of 'geometrically-decaying' kernels $\mathsf{K}_{i,j} = f(\|x_i - x_j\|)$ ($f(tx) \leq (1-\alpha)^t f(x)$), see Theorem B.5 for the formal statement of our main result. It is also noteworthy that our data structure is deterministic, and therefore handles even *adaptive* update sequences (Hardt & Woodruff, 2013; Ben-Eliezer et al., 2020; Cherapanamjeri & Nelson, 2020). This feature is important in adaptive data analysis and in the use of dynamic data structures for accelerating *path-following* iterative optimization algorithms (Brand et al., 2020), where proximity to the original gradient flow (linear) equations is crucial for convergence, hence the data structure needs to ensure the approximation guarantees hold against *any* outcome of previous iterations.

**Remark on Dynamization of "Decomposable" Problems**   A data structure problem $\mathbf{P}(D, q)$ is called *decomposable*, if a query $q$ to the *union* of two separate datasets can be recovered from the two *marginal* answers of the query on each of them separately, i.e., $\mathbf{P}(D_1 \cup D_2, q) = g(\mathbf{P}(D_1, q), \mathbf{P}(D_2, q))$ for some function $g$. A classic technique in data structures (Bentley & Saxe, 1980) asserts that decomposable data structure problems can be (partially) dynamized in a *black-box* fashion – It is possible to convert any *static* DS for $\mathbf{P}$ into a dynamic one supporting incremental updates, with an amortized update time $t_u \sim (T/N) \cdot \log(N)$, where $T$ is the preprocessing time of building the static data structure, and $N$ is the input size. We can see that Matrix-Vector multiplication over a field with row-updates to the matrix is a decomposable problem since $(A + B)q = Aq + Bq$, and so one might hope that the dynamization of static FMM/FGT methods is an immediate consequence of decomposability. This reasoning is, unfortunately, incorrect, since changing even a *single* input point $x_i \in \mathbb{R}^d$, perturbs $n$ distances, i.e., an entire *row* in the kernel matrix $\mathsf{K}$, and so the aforementioned reduction is prohibitively expensive (yields update time at least $n \gg d$ for adding/removing a point).

**Notation.**   For a vector $x$, we use $\|x\|_2$ to denote its $\ell_2$-norm, $\|x\|_1$, $\|x\|_0$ and $\|x\|_\infty$ for its $\ell_1$-norm, $\ell_0$-norm and $\ell_\infty$-norm. We use $\widetilde{O}(f)$ to denote $f \cdot \text{poly}(\log f)$. For a vector $x \in \mathbb{R}^d$ and a real number $p$, we say $x \leq p$ if $x_i \leq p$ for all $i \in [d]$. We say $x \geq p$ if there exists an $i \in [d]$ such that $x_i \geq p$. For a positive integer $n$, we use $[n]$ to denote a set $\{1, 2, \cdots, n\}$.

**Roadmap.**   In Section 2, we introduce the related research works. In Section 3, we present the important techniques used to prove our main result. In Section 4, we make a conclusion for our work.

## 2   RELATED WORK

**Structured Linear Algebra**   Multiplying an $n \times n$ matrix $M$ by an arbitrary vector $q \in \mathbb{R}^n$ generally requires $\Theta(n^2)$ time, and this is information-theoretically optimal since merely reading the entries of the matrix requires $\sim n^2$ operations. Nevertheless, if $M$ has some *structure* ($\widetilde{O}(n)$-bit description-size), one could hope for quasi-linear time for computing $M \cdot q$. Kernel matrices

$\mathsf{K}_{ij} = f(\|x_i - x_j\|)$, which are the subject of this paper, are special cases of such *geometric-analytic* structure, as their $n^2$ entries are determined by only $\sim n$ points in $\mathbb{R}^d$, i.e., $O(nd)$ bits of information. There is a rich and active body of work in *structured linear algebra*, exploring various "algebraic" structures that allow quasi-linear time matrix-vector multiplication, most of which relies on (novel) extensions of the *Fast Fourier Transform* (see (Driscoll et al., 1997; Sa et al., 2018; Chen et al., 2021) and references therein). A key difference between FMMs and the aforementioned FFT-style line of work is that the latter develops *exact* Mat-Vec algorithms, whereas FMM techniques must inevitably resort to (small) approximation, based on the *analytic* smoothness properties of the underlying function and metric space (Alman et al., 2020; 2021). This distinction makes the two lines of work mostly incomparable.

**Comparison to LSH-based KDEs**  A recent line of work due to (Charikar & Siminelakis, 2017; Backurs et al., 2018; Charikar & Siminelakis, 2019; Charikar et al., 2020; Bakshi et al., 2023) develops fast KDE data structures based on *locality-sensitive hashing* (LSH), which seems possible to be dynamized naturally (as LSH is dynamic by nature). However, this line of work is incomparable to FGT, as it solves KDE in the *low-accuracy* regime, i.e., the runtime dependence on $\varepsilon$ of these works is $\mathrm{poly}(1/\varepsilon)$ (but polynomial in $d$), as opposed to FGT ($\mathrm{poly}\log(1/\varepsilon)$ but exponential in $d$). Additionally, some work (e.g., (Charikar et al., 2020)) also needs an upper bound of the ground-truth value $\mu_\star = \mathsf{K} \cdot q$, and the efficiency of their data structure depends on $\mu_\star^{-O(1)}$, while FGT does not need any prior knowledge of $\mu_\star$.

**Kernel Methods in ML**  Kernel methods can be thought of as instance-based learners: rather than learning some fixed set of parameters corresponding to the features of their inputs, they instead "remember" the $i$-th training example $(x_i, y_i)$ and learn for it a corresponding weight $w_i$. Prediction for unlabeled inputs, i.e., those not in the training set, is treated using an application of a *similarity* function $\mathsf{K}$ (i.e., a kernel) between the unlabeled input $x'$ and each of the training-set inputs $x_i$. This framework is one of the main motivations for the development of kernel methods in ML and high-dimensional statistics (Schölkopf et al., 2002). There are two main themes of research on kernel methods in the context of machine learning: The first one is focused on understanding the expressive power and generalization of learning with kernel feature maps (Ng et al., 2002; Schölkopf et al., 2002; Shawe-Taylor & Cristianini, 2004; Rahimi & Recht, 2008; Hofmann et al., 2008; Jacot et al., 2018; Du et al., 2019; Yang et al., 2023); The second line is focused on the *computational* aspects of kernel-based algorithms (Alman et al., 2020; Brand et al., 2021; Song et al., 2021a;b; Hu et al., 2022; Alman et al., 2022; Zhang, 2022; Alman & Song, 2023; Deng et al., 2023; Gao et al., 2023b;a). We refer the reader to these references for a much more thorough overview of these lines of research and the role of kernels in ML.

## 3 Technical Overview

In Section 3.1, we review the *offline* FGT algorithm (Greengard & Rokhlin, 1987; Alman et al., 2020) and analyze the computational costs. In Section 3.2, we illustrate the technique of estimating $G(t)$ for an arbitrary target vector $t \in \mathbb{R}^d$. In Section 3.3, we explain that the data structures support the dynamic setting where the source vectors are allowed to come and leave. In Section 3.4, we describe how to extend the data structure to a more general kernel function. In Section 3.5, we show that if the source and target vectors come from a low dimensional subspace, the data structure can bypass the curse of dimension. In Section 3.6, we modify the data structure to support the scenario where the rank of data points varies across iterations.

### 3.1 Offline FGT Algorithm

We first review (Alman et al., 2020)'s offline FGT algorithm. Consider the following easier problem: given $N$ source vectors $s_1, \ldots, s_N \in \mathbb{R}^d$, and $M$ target vectors $t_1, \ldots, t_M \in \mathbb{R}^d$, estimate

$$G(t_i) = \sum_{j=1}^{N} q_j \cdot e^{-\|t_i - s_j\|_2^2/\delta}$$

for any $i \in [M]$, in quasi-linear time. Following (Greengard & Strain, 1991; Alman et al., 2020), our algorithm subdivides $B_0 = [0, 1]^d$ into smaller boxes with sides of length $L = r\sqrt{2\delta}$ parallel to

the axes, for a fixed $r \leq 1/2$, and then assign each source $s_j$ to the box $\mathcal{B}$ in which it lies and each target $t_i$ to the box $\mathcal{C}$ in which it lies. Note that there are $(1/L)^d$ boxes in total. Let $N(B)$ and $N(C)$ denote the number of non-empty source and target boxes, respectively. For each target box $\mathcal{C}$, we need to evaluate the total field due to sources in all boxes. Since each box $\mathcal{B}$ has side length $r\sqrt{2\delta}$, only a fixed number of source boxes $\mathcal{B}$ can contribute more than $\|q\|_1 \varepsilon$ to the field in a given target box $\mathcal{C}$, where $\varepsilon$ is the precision parameter. Hence, for a target vector in box $\mathcal{C}$, if we only count the contributions of the source vectors in its $(2k+1)^d$ nearest boxes where $k$ is a parameter, it will incur an error that can be upper bounded as follows:

$$\sum_{j: \|t-s_j\|_\infty \geq kr\sqrt{2\delta}} |q_j| \cdot e^{-\|t-s_j\|_2^2/\delta} \leq \|q\|_1 \cdot e^{-2r^2 k^2} \tag{1}$$

When we take $k = \log(\|q\|_1/\varepsilon)$, this error becomes $o(\varepsilon)$. For a single source vector $s_j \in \mathcal{B}$, its field $G_{s_j}(t) = q_j \cdot e^{-\|t-s_j\|^2/\delta}$ has the following Taylor expansion at $t_\mathcal{C}$ (the center of $\mathcal{C}$):

$$G_{s_j}(t) = \sum_{\beta \geq 0} \mathcal{B}_\beta(j, \mathcal{C}) \left( \frac{t - t_\mathcal{C}}{\sqrt{\delta}} \right)^\beta, \tag{2}$$

where $\beta \in \mathbb{N}^d$ is a multi-index,

$$\mathcal{B}_\beta(j, \mathcal{C}) = q_j \cdot \frac{(-1)^{\|\beta\|_1}}{\beta!} \cdot H_\beta \left( \frac{s_j - t_\mathcal{C}}{\sqrt{\delta}} \right),$$

and $H_\beta(x)$ is the multi-dimensional Hermite function indexed by $\beta$ (see Definition A.7). We can also control the truncation error of the first $p^d$ terms by $\varepsilon$ for $p = \log(\|q\|_1/\varepsilon)$ (see Lemma E.6). Then, for a fixed source box $\mathcal{B}$, the field can be approximated by

$$\sum_{\beta \leq p} C_\beta(\mathcal{B}, \mathcal{C})(\frac{t - t_\mathcal{C}}{\sqrt{\delta}})^\beta,$$

where $C_\beta(\mathcal{B}, \mathcal{C}) := \sum_{j \in \mathcal{B}} \mathcal{B}_\beta(j, \mathcal{C})$. Hence, for each query point $t$, we just need to locate its target box $\mathcal{C}$, and then $G(t)$ can be approximated by:

$$\widetilde{G}(t) = \sum_{\mathcal{B} \in \mathsf{nb}(\mathcal{C})} \sum_{\beta \leq p} C_\beta(\mathcal{B}, \mathcal{C}) \left( \frac{t - t_\mathcal{C}}{\sqrt{\delta}} \right)^\beta = \sum_{\beta \leq p} C_\beta(\mathcal{C}) \left( \frac{t - t_\mathcal{C}}{\sqrt{\delta}} \right)^\beta,$$

where $\mathsf{nb}(\mathcal{C})$ is the set of $(2k+1)^d$ nearest-neighbor of $\mathcal{C}$ and

$$C_\beta(\mathcal{C}) := \sum_{\mathcal{B} \in \mathsf{nb}(\mathcal{C})} C_\beta(\mathcal{B}, \mathcal{C}).$$

Notice that we can further pre-compute $C_\beta(\mathcal{C})$ for each target box $\mathcal{C}$ and $\beta \leq p$. Then, the running time for each target point becomes $O(p^d)$. For the preprocessing time, notice that each $C_\beta(\mathcal{B}, \mathcal{C})$ takes $O(N_\mathcal{B})$-time to compute, where $N_\mathcal{B}$ is the number of source points in $\mathcal{B}$. Fix a $\beta \leq p$. Consider the computational cost of $C_\beta(\mathcal{C})$ for all target boxes $\mathcal{C}$. Note that each source box can interact with at most $(2k+1)^d$ target boxes. Therefore, the total running time for computing $\{C_\beta(\mathcal{C}_\ell)\}_{\ell \in [N(C)]}$ is bounded by $O\left(N \cdot (2k+1)^d + M\right)$. Then, the total cost of the preprocessing is

$$O\left(N \cdot (2k+1)^d \cdot p^d + M \cdot p^d\right).$$

By taking $p = \log(\|q\|_1/\varepsilon)$ and $k \leq \log(\|q\|_1/\varepsilon)$, we get an algorithm with $\widetilde{O}_d(N + M)$-time for preprocessing and $\widetilde{O}_d(1)$-time for each target point. We note that this algorithm also supports fast computing $\mathsf{K}q$ for any $q \in \mathbb{R}^d$ and $\mathsf{K} \in \mathbb{R}^{n \times n}$ with $\mathsf{K}_{i,j} = e^{-\|s_i-s_j\|_2^2/\delta}$. Roughly speaking, for each query vector $q$, we can build this data structure, and then the $i$-th coordinate of $\mathsf{K}q$ is just $G(s_i)$, which can be computed in poly-logarithmic time. Hence, $\mathsf{K}q$ can be approximately computed in nearly-linear time with $\ell_\infty$ error at most $\varepsilon$.

**Remark 3.1.** *The kernel bandwidth $\delta > 0$ can be set using standard rules like median heuristic or cross-validation. For the box length $L = r\sqrt{2\delta}$, the parameter $r$ controls the tradeoff between computational cost and accuracy. We recommend $r = 1/2$ as it provides a good balance, and the error bound (see Eq. (1)) scales as $\exp(-2r^2 k^2)$ where $k$ is a parameter that controls the number of neighboring boxes. For the truncation parameter $p$, we set it to $p = \log(\|q\|_1/\varepsilon)$ to achieve desired accuracy $\varepsilon$ (see Lemma E.6). This parameter can be adjusted dynamically based on observed errors.*

## 3.2 ONLINE STATIC KDE DATA STRUCTURE (QUERY-ONLY)

Next, we consider the same static setting, except target queries $t \in \mathbb{R}^d$ arrive online, and the goal is to estimate $G(t)$ for an arbitrary vector in *sublinear* time. To this end, note that if $t$ is contained in a non-empty target box $\mathcal{C}_\ell$, then $G(t)$ can be approximated using pre-computed $C_\beta(\mathcal{C}_\ell)$ in poly-logarithmic time. Otherwise, we need to add a new target box $\mathcal{C}_{N(C)+1}$ for $t$ and compute $C_\beta(\mathcal{C}_{N(C)+1})$, which takes time $\sum_{\mathcal{B} \in \mathsf{nb}(\mathcal{C}_{N(C)+1})} O(N_\mathcal{B})$. However, this linear scan naïvely takes $O(N)$ time in the worst case. Indeed, looking into the coefficients $C_\beta(\mathcal{B}, \mathcal{C})$:

$$C_\beta(\mathcal{B}, \mathcal{C}) = \sum_{j \in \mathcal{B}} q_j \cdot \frac{(-1)^{\|\beta\|_1}}{\beta!} \cdot H_\beta \left( \frac{s_j - t_\mathcal{C}}{\sqrt{\delta}} \right)$$

reveals that the source vectors $s_j$ are "entangled" with $t_\mathcal{C}$, so evaluating $C_\beta(\mathcal{B}, \mathcal{C})$ brute-forcely for a new target box $\mathcal{C}$, incurs a linear scan of all source vectors in $\mathcal{B}$. To "disentangle" $s_j$ and $t_\mathcal{C}$, we use the Taylor series of Hermite function (Eq. (5)):

$$
\begin{aligned}
H_\beta \left( \frac{s_j - t_\mathcal{C}}{\sqrt{\delta}} \right) &= H_\beta \left( \frac{s_j - s_\mathcal{B}}{\sqrt{\delta}} + \frac{s_\mathcal{B} - t_\mathcal{C}}{\sqrt{\delta}} \right) \\
&= \sum_{\alpha \geq 0} \frac{(-1)^{\|\alpha\|_1}}{\alpha!} \left( \frac{s_j - s_\mathcal{B}}{\sqrt{\delta}} \right)^\alpha H_{\alpha+\beta} \left( \frac{s_\mathcal{B} - t_\mathcal{C}}{\sqrt{\delta}} \right),
\end{aligned}
$$

where $s_\mathcal{B}$ denotes the center of the source box $\mathcal{B}$. Hence, $C_\beta(\mathcal{B}, \mathcal{C})$ can be re-written as:

$$
\begin{aligned}
C_\beta(\mathcal{B}, \mathcal{C}) &= \sum_{j \in \mathcal{B}} q_j g(\beta) \sum_{\alpha \geq 0} g(\alpha) \left( \frac{s_j - s_\mathcal{B}}{\sqrt{\delta}} \right)^\alpha H_{\alpha+\beta} \left( \frac{s_\mathcal{B} - t_\mathcal{C}}{\sqrt{\delta}} \right) \\
&= g(\beta) \sum_{\alpha \geq 0} A_\alpha(\mathcal{B}) H_{\alpha+\beta} \left( \frac{s_\mathcal{B} - t_\mathcal{C}}{\sqrt{\delta}} \right),
\end{aligned}
$$

where $g(x) = (-1)^{\|x\|_1} / x!$ and

$$A_\alpha(\mathcal{B}) := \sum_{j \in \mathcal{B}} q_j g(\alpha) \left( \frac{s_j - s_\mathcal{B}}{\sqrt{\delta}} \right)^\alpha. \tag{3}$$

Now, $A_\alpha(\mathcal{B})$ does not rely on the target box and can be pre-computed, hence we can compute $C_\beta(\mathcal{B}, \mathcal{C})$ without going over each source vector. However, there is a price for this conversion, namely, that now $C_\beta(\mathcal{B}, \mathcal{C})$ involves summing over all $\alpha \geq 0$, so we need to somehow truncate this series while controlling the overall truncation error for $G(t)$, which appears difficult to achieve. To this end, we observe that this two-step approximation is equivalent to first forming a truncated Hermite series of $e^{\|t - s_j\|_2^2 / \delta}$ at the center of the source box $s_\mathcal{B}$, and then transforming all Hermite expansions into Taylor expansions at the center of a *target* box $t_\mathcal{C}$. More formally, the Hermite approximation of $G(t)$ is

$$G(t) = \sum_\mathcal{B} \sum_{\alpha \leq p} (-1)^{\|\alpha\|_1} A_\alpha(\mathcal{B}) H_\alpha \left( \frac{t - s_\mathcal{B}}{\sqrt{\delta}} \right) + \mathrm{Err}_H(p),$$

where $|\mathrm{Err}_H(p)| \leq \varepsilon$ (see Lemma E.2). Hence, we can Taylor-expand each $H_\alpha$ at $t_\mathcal{C}$ and get that:

$G(t) = \sum_{\beta \leq p} C_\beta(\mathcal{C}) \left( \frac{t - t_\mathcal{C}}{\sqrt{\delta}} \right)^\beta + \mathrm{Err}_T(p) + \mathrm{Err}_H(p)$, where $|\mathrm{Err}_H(p)| + |\mathrm{Err}_T(p)| \leq \varepsilon$, (for the formal argument, see Lemma E.5).

**Remark 3.2.** *The original FGT paper contains a flaw in the error estimation, which was partially fixed in (Baxter & Roussos, 2002) for the Hermite expansion. Later, (Lee et al., 2005) corrected the error in both Hermite and Taylor expansions. However, their proofs are brief and use different notations that are adapted for their dual-tree algorithm. We provide more detailed and user-friendly proofs for the correct error estimations in Section E. We believe that they are of independent interest to the community.*

This means that, at preprocessing time, it suffices to compute $A_\alpha(\mathcal{B})$ for all source boxes and all $\alpha \leq p$, which takes

$$\sum_{k \in [N(B)]} O\left(p^d \cdot N_{\mathcal{B}_k}\right) = O\left(p^d \cdot N\right) = \widetilde{O}_d(N).$$

time. Then, at query time, given an arbitrary query vector $t$ in a target box $\mathcal{C}$, we compute

$$C_\beta(\mathcal{C}) = h(\beta) \sum_{\mathcal{B} \in \mathsf{nb}(\mathcal{C})} \sum_{\alpha \leq p} A_\alpha(\mathcal{B}) H_{\alpha+\beta}\left(\frac{s_\mathcal{B} - t_\mathcal{C}}{\sqrt{\delta}}\right),$$

which takes

$$O\left(d \cdot p^d \cdot (2k+1)^d\right) = \mathrm{poly}\log(n)$$

time, so long as $d = O(1)$ and $\varepsilon = n^{-O(1)}$.

## 3.3 DYNAMIZATION

Given our (static) representation of points from the last paragraph, dynamizing the above static KDE data structure now becomes simple. Suppose we add a source vector $s$ in the source box $\mathcal{B}$. We first update the intermediate variables $A_\alpha(\mathcal{B}), \alpha \leq p$, which takes $O(p^d)$ time. So long as the $\ell_1$-norm of the updated charge-vector $q$ remains polynomial in the norm of the previously maintained vector, namely

$$\sqrt{\log(\|q^{\mathrm{new}}\|_1)} > \log(\|q\|_1),$$

we show that one source box can only affect $(2k+1)^d$ nearest target box $\mathcal{C}$; otherwise, when the change is super-polynomial, we rebuild the data structure, but this cost is amortized away. Hence, we only need to update $C_\beta(\mathcal{C})$ for those $\mathcal{C} \in \mathsf{nb}(\mathcal{B})$. Notice that each $C_\beta(\mathcal{B}, \mathcal{C})$ can be updated in $O_d(1)$ time, so each affected $C_\beta(\mathcal{C})$ can also be updated in $O_d(1)$ time. Hence, adding a source vector can be done in time $O((2k+1)^d p^d) = \widetilde{O}_d(1)$ as before. Deleting a source vector follows from a similar procedure.

## 3.4 GENERALIZATION TO FAST-DECAYING KERNELS

We briefly explain how the dynamic FGT data structure generalizes to more general kernel functions $\mathsf{K}(s,t) = f(\|s-t\|_2)$ where $f$ satisfies the 3 properties in Definition 3.3 below.

**Definition 3.3** (Properties of general kernel function, (Alman et al., 2020)). *We define the following properties of the function $f : \mathbb{R} \to \mathbb{R}_+$:*

- **P1:** *$f$ is non-increasing, i.e., $f(x) \leq f(y)$ when $x \geq y$.*

- **P2:** *$f$ is decreasing fast, i.e., $f(\Theta(\log(1/\varepsilon))) \leq \varepsilon$.*

- **P3:** *$f$'s Hermite expansion and Taylor expansion are truncateable: the truncation error of the first $(\log^d(1/\varepsilon))$ terms in the Hermite and Taylor expansion of $\mathsf{K}$ is at most $\varepsilon$.*

**Remark 3.4.** *There are many widely-used kernels that satisfy the properties of general kernel function (Definition 3.3) such as:*

- *inverse polynomial kernels: $K(x,y) = 1/\|x-y\|_2^c$ for constant $c > 0$,*

- *exponential kernel: $K(x,y) = \exp(-\|x-y\|_2)$,*

- *inverse multiquadric kernel: $K(x,y) = 1/\sqrt{\|x-y\|_2^2 + c}$ (Micchelli, 1984; Martinsson, 2012), and*

- *rational quadratic kernel: $K(x,y) = 1/(1 + \|x-y\|_2^2/\alpha)$ for $\alpha > 0$.*

*The key insight is that these kernels' fast decay allows truncation of distant interactions, while their smoothness enables efficient local approximations via series expansions. This broader applicability significantly extends the practical utility of our dynamic data structure.*

In the general case, $G_f(t) = \sum_{\mathcal{B}} \sum_{j \in \mathcal{B}} q_j \mathsf{K}(s_j, t)$. Similar to the Gaussian kernel case, we can first show that only near boxes matter:

$$\sum_{j:\|t-s_j\|_\infty \geq kr} |q_j| \cdot f(\|s - t\|_2) \leq \varepsilon$$

by the fast-decreasing property (**P2**) in Definition 3.3 of $f$ and taking $k = O(\log(\|q\|_1/\varepsilon))$[1]. Then, we can follow the same "decoupling" approach as the Gaussian kernel case to first Hermite expand $G_f(t)$ at the center of each source box and then Taylor expands each Hermite function at the center of the target box. In this way, we can show that

$$G_f(t) \approx \sum_{\beta \leq p} C_{f,\beta}(\mathcal{C}) \left( \frac{t - t_\mathcal{C}}{\sqrt{\delta}} \right)^\beta,$$

where $C_{f,\beta}(\mathcal{C}) = c_\beta \sum_{\mathcal{B} \in \mathsf{nb}(\mathcal{C})} \sum_{\alpha \leq p} A_{f,\alpha}(\mathcal{B}) H_{\alpha+\beta} \left( \frac{s_\mathcal{B} - t_\mathcal{C}}{\sqrt{\delta}} \right)$, and the approximation error can be bounded since $f$ is truncateable. $A_{f,\alpha}(\mathcal{B})$ depends on the kernel function $f$ and can be pre-computed in the preprocessing. Then, each $C_{f,\beta}(\mathcal{C})$ can be computed in poly-logarithmic time. Hence, $G(t)$ can be approximately computed in poly-logarithmic time for any target vector $t$.

### 3.5 HANDLING POINTS FROM LOW-DIMENSIONAL STATIC SPACES

In many practical problems, the data lies in a low dimensional subspace of $\mathbb{R}^d$. We can first project the data into this subspace and then perform FGT on $\mathbb{R}^w$, where $w$ is the rank. The following lemma shows that FGT can be performed on the projections of the data.

**Lemma 3.5** (Hermite projection lemma in low-dimensional space, informal version of Lemma F.3). *Given $\mathcal{B} \in \mathbb{R}^{d \times w}$ that defines a $w$-dimensional subspace of $\mathbb{R}^d$, let $\mathcal{B}^\top \mathcal{B} = U \Lambda U^\top \in \mathbb{R}^{w \times w}$ denote the spectral decomposition where $U \in \mathbb{R}^{w \times w}$ and a diagonal matrix $\Lambda \in \mathbb{R}^{w \times w}$. We define $\mathsf{P} := \Lambda^{-1/2} U^{-1} \mathcal{B}^\top \in \mathbb{R}^{w \times d}$. Then we have for any $t, s \in \mathbb{R}^d$ from subspace $\mathcal{B}$, the following equation holds*

$$e^{-\|t-s\|_2^2/\delta} = \sum_{\alpha \geq 0} \frac{(\sqrt{1/\delta}\mathsf{P}(t-s))^\alpha}{\alpha!} h_\alpha(\sqrt{1/\delta}\mathsf{P}(t-s)).$$

By Lemma 3.5, it suffices to divide $\mathbb{R}^w$ instead of $\mathbb{R}^d$ into boxes and conduct Hermite expansion and Taylor expansion on the low-dimensional subspace. More specifically, given the initial source points, we can compute $\mathsf{P}$ by SVD or QR decomposition in $N \cdot w^{\omega-1}$-time[2], which is of smaller order than the FGT's preprocessing time[3]. Then, we can project each point $s_i \in \mathbb{R}^d$ to $x_i := \mathsf{P}s_i \in \mathbb{R}^w$ for $i \in [N]$. The remaining procedure in preprocessing is the same as before, but directly working on the low-dimensional sources $\{x_1, \ldots, x_N\}$. In the query phase, consider a target point $t$ in the subspace. We are supposed to compute $G(t) \approx \sum_{\mathcal{B}} \sum_{j \in \mathcal{B}} q_j \cdot e^{-\|t-s_j\|_2^2/\delta}$. By Lemma 3.5, we know that $G(t) \approx \sum_{\beta \leq p} C_\beta(\mathcal{C}) \left( \frac{\mathsf{P}(t-t_\mathcal{C})}{\sqrt{\delta}} \right)^\beta = \sum_{\beta \leq p} C_\beta(\mathcal{C}) \left( \frac{y - y_\mathcal{C}}{\sqrt{\delta}} \right)^\beta$, where $\mathcal{C}$ is the target box that contains $t$, $y = \mathsf{P}t$ and $y_\mathcal{C} = \mathsf{P}t_\mathcal{C}$ projected points. Moreover, for each $\beta \leq p$ and target box $\mathcal{C}$, we have

$$C_\beta(\mathcal{C}) = \frac{(-1)^{\|\beta\|_1}}{\beta!} \sum_{\mathcal{B}} \sum_{\alpha \leq p} A_\alpha(\mathcal{B}) H_{\alpha+\beta} \left( \frac{\mathsf{P}(s_\mathcal{B} - t_\mathcal{C})}{\sqrt{\delta}} \right)$$

$$= \frac{(-1)^{\|\beta\|_1}}{\beta!} \sum_{\mathcal{B}} \sum_{\alpha \leq p} A_\alpha(\mathcal{B}) H_{\alpha+\beta} \left( \frac{x_\mathcal{B} - y_\mathcal{C}}{\sqrt{\delta}} \right).$$

---

[1] Indeed, by property **P2**, $f(\Theta(\log(1/\varepsilon'))) \leq \varepsilon'$. Taking $\varepsilon' := \varepsilon/\|q\|_1$, we get that $f(\|s-t\|_2) \leq \varepsilon/\|q\|_1$. Hence, the summation is at most $\varepsilon$.

[2] $\omega \approx 2.372$ is the fast matrix multiplication time exponent.

[3] In practice, we can run numerical algorithms such as randomized SVD that are very fast for low-rank matrices.

Similarly, for each $\alpha \leq p$ and source box $\mathcal{B}$,

$$A_\alpha(\mathcal{B}) = \frac{(-1)^{\|\alpha\|_1}}{\alpha!} \sum_{j \in \mathcal{B}} q_j \cdot \left( \frac{x_j - x_\mathcal{B}}{\sqrt{\delta}} \right)^\alpha.$$

Therefore, each query is equivalent to being conducted in a $w$-dimensional space using our data structure, which takes $\log^{O(w)}(\|q\|_1/\varepsilon)$-time. The update can be done in a similar way in the low-dimensional space using the procedure described in Section 3.3. Hence, each update (insertion or deletion) takes $\log^{O(w)}(\|q\|_1/\varepsilon)$.

### 3.6 HANDLING POINTS FROM LOW-DIMENSIONAL DYNAMIC SPACES

We note that when we add a new source point to the data structure, the intrinsic rank of the data might change by 1 when the point is not in the subspace. For an inserting source point $s$, consider the rank-increasing case, i.e., $(I - \mathsf{P})s \neq 0$. Then, this new source point contributes to one new basis $u := \frac{(I - \mathsf{P})s}{\|(I-\mathsf{P})s\|_2}$. Also, we can update the projection matrix $\mathsf{P}$ by $[\mathsf{P} \quad u] \in \mathbb{R}^{(w+1) \times d}$. However, as the subspace is changed, we need to maintain the intermediate variables $A_\alpha(\mathcal{B}), C_\beta(\mathcal{C})$. It is easy to observe that for the original projected source and target points or boxes, they can easily be "lifted" to the new subspace by setting zero to the $(w + 1)$-th coordinate. We show how to update $A_\alpha(\mathcal{B})$ efficiently. For each source box $\mathcal{B}$ and $\alpha \leq p$, we have

$$A^{\mathsf{new}}_{(\alpha,0)}(\mathcal{B}) = \frac{(-1)^{\|\alpha\|_1} \cdot (-1)^i}{\alpha! \cdot i!} \sum_{j \in \mathcal{B}} q_j \cdot \left( \frac{x'_j - x'_\mathcal{B}}{\sqrt{\delta}} \right)^{(\alpha,i)} = A_\alpha(\mathcal{B}),$$

where $x'_j$ denotes the lifted point. And $A^{\mathsf{new}}_{(\alpha,1)}(\mathcal{B}) = 0$ for all $i > 0$. Similarly, for each target box $\mathcal{C}$,

$$C^{\mathsf{new}}_{(\beta,i)}(\mathcal{C}) = \frac{(-1)^{\|\beta\|_1}(-1)^i}{\beta! i!} \cdot \sum_\mathcal{B} \sum_{\alpha \leq p} \sum_{j=0}^p A^{\mathsf{new}}_{(\alpha,j)}(\mathcal{B}) H_{(\alpha+\beta, i+j)} \left( \frac{x'_\mathcal{B} - y'_\mathcal{C}}{\sqrt{\delta}} \right)$$

$$= \frac{(-1)^{\|\beta\|_1}(-1)^i}{\beta! i!} \cdot \sum_\mathcal{B} \sum_{\alpha \leq p} A_\alpha(\mathcal{B}) H_{\alpha+\beta} \left( \frac{x_\mathcal{B} - y_\mathcal{C}}{\sqrt{\delta}} \right) \cdot h_i(0)$$

$$= \frac{(-1)^i}{i!} \cdot C_\beta(\mathcal{C}).$$

Therefore, by enumerating all boxes $\mathcal{B}, \mathcal{C}$ and indices $\alpha, \beta \leq p$, we can compute $A^{\mathsf{new}}_{(\alpha,0)}(\mathcal{B})$ and $C^{\mathsf{new}}_{(\beta,i)}(\mathcal{C})$ in $\log^{O(w)}(\|q\|_1/\varepsilon)$-time. Then, we just follow the static subspace insertion procedure to insert the new source point $s$. In this way, we obtain a data structure that can handle dynamic low-rank subspaces.

## 4 CONCLUSION AND FUTURE DIRECTIONS

In this paper, we study the Fast Gaussian Transform (FGT) in a dynamic setting and propose a dynamic data structure to maintain the source vectors that support very fast kernel density estimation, Mat-Vec queries ($\mathsf{K} \cdot q$), as well as updating the source vectors. We further show that the efficiency of our algorithm can be improved when the data points lie in a low-dimensional subspace. Our results are especially valuable when FGT is used in real-world applications with rapidly-evolving datasets, e.g., online regression, federated learning, etc.

One open problem in this direction is, can we compute $\mathsf{K}q$ in $O(N) + \log^{O(d)}(N/\varepsilon)$ time? Currently, it takes $N \log^{O(d)}(N/\varepsilon)$ time even in the static setting. The lower bounds in (Alman et al., 2020) indicate that this improvement is impossible for some "bad" kernels $\mathsf{K}$ which are very non-smooth. It remains open when $\mathsf{K}$ is a Gaussian-like kernel. It might be helpful to apply more complicated geometric data structures to maintain the interactions between data points. Another open problem is, can we fast compute Mat-Vec product or KDE for slowly-decaying kernels? The main difficulty is the current FMM techniques cannot achieve high accuracy when the kernel decays slowly. New techniques might be required to resolve this problem.

## ETHIC STATEMENT

This paper does not involve human subjects, personally identifiable data, or sensitive applications. We do not foresee direct ethical risks. We follow the ICLR Code of Ethics and affirm that all aspects of this research comply with the principles of fairness, transparency, and integrity.

## REPRODUCIBILITY STATEMENT

We ensure reproducibility of our theoretical results by including all formal assumptions, definitions, and complete proofs in the appendix. The main text states each theorem clearly and refers to the detailed proofs. No external data or software is required.

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

# Appendix

**Roadmap.** In Section A, we provide several notations and definitions about the Fast Multipole Method. In Section B, we present the formal statement of our main result. In Section C, we present our data-structures and algorithms. In Section D, we provide a complete and full for our results. In Section E, we prove several lemmas to control the error. In Section F, we generalize our results to low dimension subspace setting.

## A    PRELIMINARIES

We first give a quick overview of the high-level ideas of FMM in Section A.1. In Section A.2, we provide a complete description and proof of correctness for the fast Gaussian transform, where the kernel function is the Gaussian kernel. Although a number of researchers have used FMM in the past, most of the previous papers about FMM either focus on low-dimensional or low-error cases. We therefore focus on the superconstant-error, high dimensional case, and carefully analyze the joint dependence on $\varepsilon$ and $d$. We believe that our presentation of the original proof in Section A.2 is thus of independent interest to the community.

### A.1    FMM BACKGROUND

We begin with a description of high-level ideas of the Fast Multipole Method (FMM). Let $\mathsf{K} : \mathbb{R}^d \times \mathbb{R}^d \to \mathbb{R}_+$ denote a kernel function. The inputs to the FMM are $N$ sources $s_1, s_2, \cdots, s_N \in \mathbb{R}^d$ and $M$ targets $t_1, t_2, \cdots, t_M$. For each $i \in [N]$, source $s_i$ has associated 'strength' $q_i$. Suppose all sources are in a 'box' $\mathcal{B}$ and all the targets are in a 'box' $\mathcal{C}$. The goal is to evaluate

$$u_j = \sum_{i=1}^{N} \mathsf{K}(s_i, t_j) q_i, \quad \forall j \in [M]$$

Intuitively, if $\mathsf{K}$ has some nice property (e.g. smooth), we can hope to approximate $\mathsf{K}$ in the following sense:

$$\mathsf{K}(s, t) \approx \sum_{p=0}^{P-1} B_p(s) \cdot C_p(t), \quad s \in \mathcal{B}, t \in \mathcal{C}$$

for some functions $B_p, C_p : \mathbb{R}^d \to \mathbb{R}$, where $P$ is a small positive integer, usually called the *interaction rank* in the literature (Corona et al., 2015; Martinsson, 2019).

Now, we can construct $u_i$ in two steps:

$$v_p = \sum_{i \in \mathcal{B}} B_p(s_i) q_i, \quad \forall p = 0, 1, \cdots, P-1,$$

and

$$\widetilde{u}_j = \sum_{p=0}^{P-1} C_p(t_j) v_p, \quad \forall i \in [M].$$

Intuitively, as long as $\mathcal{B}$ and $\mathcal{C}$ are well-separated, then $\widetilde{u}_j$ is very good estimation to $u_j$ even for small $P$, i.e., $|\widetilde{u}_j - u_j| < \varepsilon$.

Recall that, at the beginning of this section, we assumed that all the sources are in the the same box $\mathcal{B}$ and $\mathcal{C}$. This is not true in general. To deal with this, we can discretize the continuous space into a batch of boxes $\mathcal{B}_1, \mathcal{B}_2, \cdots$ and $\mathcal{C}_1, \mathcal{C}_2, \cdots$. For a box $\mathcal{B}_{l_1}$ and a box $\mathcal{C}_{l_2}$, if they are very far apart, then the interaction between points within them is small, and we can ignore it. If the two boxes are close, then we deal with them efficiently by truncating the high order expansion terms in $\mathsf{K}$ (only keeping the first $\log^{O(d)}(1/\varepsilon)$ terms). For each box, we will see that the number of nearby relevant boxes is at most $\log^{O(d)}(1/\varepsilon)$.

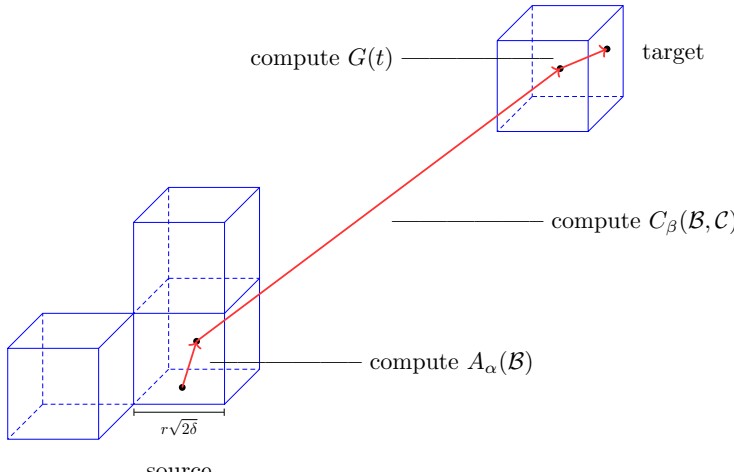

Figure 1: An illustration of the source-target boxing our data structure maintains in high dimensional space, using the "hybrid" of Taylor-Hermite expansions.

## A.2 FAST GAUSSIAN TRANSFORM

Given $N$ vectors $s_1, \cdots s_N \in \mathbb{R}^d$, $M$ vectors $t_1, \cdots, t_M \in \mathbb{R}^d$ and a strength vector $q \in \mathbb{R}^n$, Greengard and Strain (Greengard & Strain, 1991) provided a fast algorithm for evaluating discrete Gauss transform

$$G(t_i) = \sum_{j=1}^{N} q_j e^{-\|t_i - s_j\|^2/\delta}$$

for all $i \in [M]$ in $O(M+N)$ time. In this section, we re-prove the algorithm described in (Greengard & Strain, 1991), and determine the exact dependence on $\varepsilon$ and $d$ in the running time.

Without loss of generality, we can assume that all the sources $s_j$ and targets are belonging to the unit box $\mathcal{B}_0 = [0,1]^d$. The reason is, if not, we can shift the origin and rescaling $\delta$.

Let $t$ and $s$ lie in $d$-dimensional Euclidean space $\mathbb{R}^d$, and consider the Gaussian

$$e^{-\|t-s\|_2^2} = e^{-\sum_{i=1}^{d}(t_i - s_i)^2}$$

We begin with some definitions. One important tool we use is the Hermite polynomial, which is a well-known class of orthogonal polynomials with respect to Gaussian measure and widely used in analyzing Gaussian kernels.

**Definition A.1** (One-dimensional Hermite polynomial, (Hermite, 1864)). *The Hermite polynomials $\widetilde{h}_n : \mathbb{R} \to \mathbb{R}$ is defined as follows*

$$\widetilde{h}_n(t) = (-1)^n e^{t^2} \frac{\mathrm{d}^n}{\mathrm{d}t} e^{-t^2}$$

The first few Hermite polynomials are:

$$\widetilde{h}_1(t) = 2t, \ \widetilde{h}_2(t) = 4t^2 - 2, \ \widetilde{h}_3(t) = 8t^3 - 12t, \ \cdots$$

**Definition A.2** (One-dimensional Hermite function, (Hermite, 1864)). *The Hermite functions $h_n : \mathbb{R} \to \mathbb{R}$ is defined as follows*

$$h_n(t) = e^{-t^2}\widetilde{h}_n(t) = (-1)^n \frac{\mathrm{d}^n}{\mathrm{d}t} e^{-t^2}$$

We use the following Fact to simplify $e^{-(t-s)^2/\delta}$.

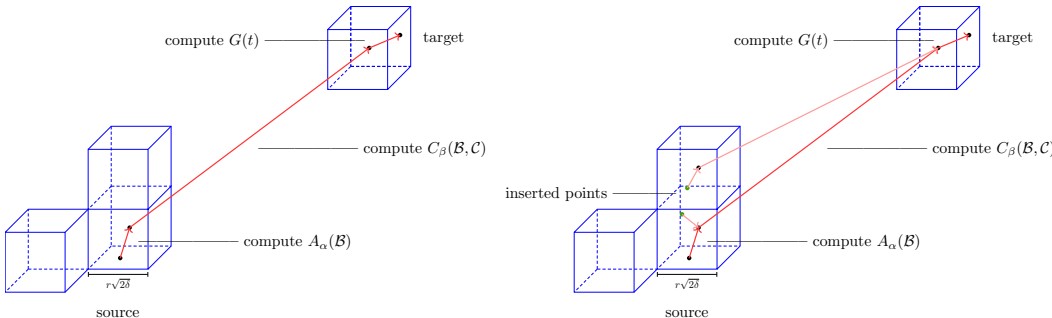

Figure 2: An illustration of inserting two source points with corresponding interactions to the data structure.

**Fact A.3.** *For $s_0 \in \mathbb{R}$ and $\delta > 0$, we have*

$$e^{-(t-s)^2/\delta} = \sum_{n=0}^{\infty} \frac{1}{n!} \cdot \left(\frac{s-s_0}{\sqrt{\delta}}\right)^n \cdot h_n\left(\frac{t-s_0}{\sqrt{\delta}}\right)$$

*and*

$$e^{-(t-s)^2/\delta} = e^{-(t-s_0)^2/\delta} \sum_{n=0}^{\infty} \frac{1}{n!} \cdot \left(\frac{s-s_0}{\sqrt{\delta}}\right)^n \cdot \widetilde{h}_n\left(\frac{t-s_0}{\sqrt{\delta}}\right).$$

**Lemma A.4** (Cramer's inequality for one-dimensional, (Hille, 1926)). *For any $K < 1.09$,*

$$|\widetilde{h}_n(t)| \leq K 2^{n/2}\sqrt{n!}e^{t^2/2}.$$

Using Cramer's inequality (Lemma A.4), we have the following standard bound.

**Lemma A.5.** *For any constant $K < 1.09$, we have*

$$|h_n(t)| \leq K \cdot 2^{n/2} \cdot \sqrt{n!} \cdot e^{-t^2/2}.$$

Next, we will extend the above definitions and observations to the high dimensional case. To simplify the discussion, we define multi-index notation. A multi-index $\alpha = (\alpha_1, \alpha_2, \cdots, \alpha_d)$ is a $d$-tuple of nonnegative integers, playing the role of a multi-dimensional index. For any multi-index $\alpha \in \mathbb{R}^d$ and any $t \in \mathbb{R}^t$, we write

$$\alpha! = \prod_{i=1}^{d}(\alpha_i!), \quad t^\alpha = \prod_{i=1}^{d}t_i^{\alpha_i}, \quad D^\alpha = \partial_1^{\alpha_1}\partial_2^{\alpha_2}\cdots\partial_d^{\alpha_d}.$$

where $\partial_i$ is the differential operator with respect to the $i$-th coordinate in $\mathbb{R}^d$. For integer $p$, we say $\alpha \leq p$ if $\alpha_i \leq p, \forall i \in [d]$; and we say $\alpha \geq p$ if $\alpha_i \geq p, \exists i \in [d]$. We use these definitions to guarantee that $\{\alpha \leq p\} \cup \{\alpha \geq p\} = \mathbb{N}^d$.

We can now define multi-dimensional Hermite polynomial:

**Definition A.6** (Multi-dimensional Hermite polynomial, (Hermite, 1864)). *We define function $\widetilde{H}_\alpha : \mathbb{R}^d \to \mathbb{R}$ as follows:*

$$\widetilde{H}_\alpha(t) = \prod_{i=1}^{d}\widetilde{h}_{\alpha_i}(t_i).$$

**Definition A.7** (Multi-dimensional Hermite function, (Hermite, 1864)). *We define function $H_\alpha : \mathbb{R}^d \to \mathbb{R}$ as follows:*

$$H_\alpha(t) = \prod_{i=1}^{d}h_{\alpha_i}(t_i).$$

*It is easy to see that $H_\alpha(t) = e^{-\|t\|_2^2} \cdot \widetilde{H}_\alpha(t)$*

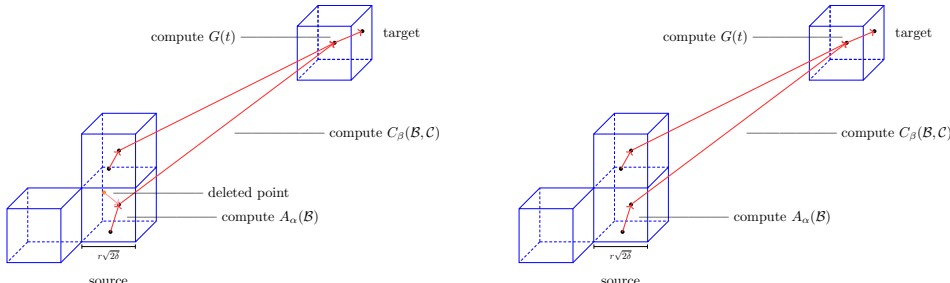

Figure 3: An illustration of deleting a source point from the data structure.

The Hermite expansion of a Gaussian in $\mathbb{R}^d$ is

$$e^{-\|t-s\|_2^2} = \sum_{\alpha \geq 0} \frac{(t-s_0)^\alpha}{\alpha!} h_\alpha(s-s_0). \tag{4}$$

Cramer's inequality generalizes to

**Lemma A.8** (Cramer's inequality for multi-dimensional case, (Greengard & Strain, 1991; Alman et al., 2020)). *Let $K < (1.09)^d$, then*

$$|\widetilde{H}_\alpha(t)| \leq K \cdot e^{\|t\|_2^2/2} \cdot 2^{\|\alpha\|_1/2} \cdot \sqrt{\alpha!}$$

*and*

$$|H_\alpha(t)| \leq K \cdot e^{-\|t\|_2^2/2} \cdot 2^{\|\alpha\|_1/2} \cdot \sqrt{\alpha!}.$$

The Taylor series of $H_\alpha$ is

$$H_\alpha(t) = \sum_{\beta \geq 0} \frac{(t-t_0)^\beta}{\beta!} (-1)^{\|\beta\|_1} H_{\alpha+\beta}(t_0). \tag{5}$$

# B OUR RESULT

## B.1 PROPERTIES OF KERNEL FUNCTION

(Alman et al., 2020) identified the three key properties of kernel functions $\mathsf{K}(s,t) = f(\|s-t\|_2)$ which allow sub-quadratic matrix-vector multiplication via the fast Multipole method. Our dynamic algorithm will work for any kernel satisfying these properties.

**Definition B.1** (Properties of general kernel function, restatement of Definition 3.3, (Alman et al., 2020)). *We define the following properties of the function $f : \mathbb{R} \to \mathbb{R}_+$:*

- **P1:** *$f$ is non-increasing, i.e., $f(x) \leq f(y)$ when $x \geq y$.*

- **P2:** *$f$ is decreasing fast, i.e., $f(\Theta(\log(1/\varepsilon))) \leq \varepsilon$.*

- **P3:** *$f$'s Hermite expansion and Taylor expansion are truncateable: the truncation error of the first $(\log^d(1/\varepsilon))$ terms in the Hermite and Taylor expansion of $\mathsf{K}$ is at most $\varepsilon$.*

**Remark B.2.** *We note that **P3** can be replaced with the following more general property:*

- **P4:** $\mathsf{K} : \mathbb{R}^d \times \mathbb{R}^d \to \mathbb{R}$ *is $\{\phi_\alpha\}_{\alpha \in \mathbb{N}^d}$-expansionable: there exist constants $c_\alpha$ that only depend on $\alpha \in \mathbb{N}^d$ and functions $\phi_\alpha : \mathbb{R}^d \to \mathbb{R}$ such that*

$$\mathsf{K}(s,t) = \sum_{\alpha \in \mathbb{N}^d} c_\alpha \cdot (s-s_0)^\alpha \cdot \phi_\alpha(t-s_0)$$

*for any $s_0 \in \mathbb{R}^d$ and $s$ close to $s_0$. Moreover, the truncation error of the first $(\log^d(1/\varepsilon))$ terms is $\leq \varepsilon$.*

---

**Algorithm 1** Informal version of Algorithm 2, 3, 4 and 5.

---

1: **data structure** DYNAMICFGT                           ▷ Theorem B.5
2:    **members**
3:       $A_\alpha(\mathcal{B}_k), k \in [N(B)], \alpha \le p$
4:       $C_\beta(\mathcal{C}_k), k \in [N(C)], \beta \le p$
5:       $t_{\mathcal{C}_k}, k \in [N(C)]$
6:       $s_{\mathcal{B}_k}, k \in [N(B)]$
7:    **end members**

8:    **procedure** UPDATE$(s \in \mathbb{R}^d, q \in \mathbb{R})$           ▷ Informal version of Algorithm 4 and 5
9:       Find the box $s \in \mathcal{B}_k$
10:      Update $A_\alpha(\mathcal{B}_k)$ for all $\alpha \le p$
11:      Find $(2k+1)^d$ nearest target boxes to $\mathcal{B}_k$, denote by $\mathsf{nb}(\mathcal{B}_k)$      ▷ $k \le \log(\|q\|_1/\varepsilon)$
12:      **for** $\mathcal{C}_l \in \mathsf{nb}(\mathcal{B}_k)$ **do**
13:         Update $C_\beta(\mathcal{C}_l)$ for all $\beta \le p$
14:      **end for**
15:   **end procedure**

16:   **procedure** KDE-QUERY$(t \in \mathbb{R}^d)$              ▷ Informal version of Algorithm 3
17:      Find the box $t \in \mathcal{C}_k$
18:      $\widetilde{G}(t) \leftarrow \sum_{\beta \le p} C_\beta(\mathcal{C}_k)((t - t_{\mathcal{C}_k})/\sqrt{\delta})^\beta$
19:   **end procedure**
20: **end data structure**

---

**Remark B.3.** *Two examples of kernels that satisfy Properties 1 and 2 are:*

- $\mathsf{K}(s,t) = e^{-\alpha\|s-t\|^2}$ *for any* $\alpha \in \mathbb{R}_+$.

- $\mathsf{K}(s,t) = e^{-\alpha\|s-t\|^{2p}}$ *for any* $p \in \mathbb{N}_+$.

## B.2 DYNAMIC FGT

In this section, we present our main result. We first define the dynamic density-estimation maintenance problem with respect to the Gaussian kernel.

**Definition B.4** (Dynamic FGT Problem). *We wish to design a data-structure that efficiently supports any sequence of the following operations:*

- INIT$(S \subset \mathbb{R}^d, q \in \mathbb{R}^{|S|}, \varepsilon \in \mathbb{R})$ *Let* $N = |S|$. *The data structure is given* $N$ *source points* $s_1, \cdots, s_N \in \mathbb{R}^d$ *with their charge* $q_1, \cdots, q_N \in \mathbb{R}$.

- INSERT$(s \in \mathbb{R}^d, \ q_s \in \mathbb{R})$ *Add the source point* $s$ *with its charge* $q_s$ *to the point set* $S$.

- DELETE$(s \in \mathbb{R}^d)$ *Delete* $s$ *(and its charge* $q_s$*) from the point set* $S$.

- KDE-QUERY$(t \in \mathbb{R}^d)$ *Output* $\widetilde{G}$ *such that* $G(t) - \varepsilon \le \widetilde{G} \le G(t) + \varepsilon$.

The main result of this paper is a fully-dynamic data structure supporting all of the above operations in *polylogarithmic* time:

**Theorem B.5** (Dynamic FGT Data Structure). *Given* $N$ *vectors* $S = \{s_1, \cdots, s_N\} \subset \mathbb{R}^d$, *a number* $\delta > 0$, *and a vector* $q \in \mathbb{R}^N$, *let* $G : \mathbb{R}^d \to \mathbb{R}$ *be defined as* $G(t) = \sum_{i=1}^N q_i \cdot \mathsf{K}(s_i, t)$ *denote the kernel-density of* $t$ *with respect to* $S$, *where* $\mathsf{K}(s_i, t) = f(\|s_i - t\|_2)$ *for* $f$ *satisfying the properties in Definition 3.3 . There is a dynamic data structure that supports the following operations:*

- INIT() *(Algorithm 2) Preprocess in* $N \cdot \log^{O(d)}(\|q\|_1/\varepsilon)$ *time.*

- KDE-QUERY$(t \in \mathbb{R}^d)$ *(Algorithm 3) Output* $\widetilde{G}$ *such that* $G(t) - \varepsilon \le \widetilde{G} \le G(t) + \varepsilon$ *in* $\log^{O(d)}(\|q\|_1/\varepsilon)$ *time.*

- INSERT$(s \in \mathbb{R}^d, q_s \in \mathbb{R})$ *(Algorithm 4) For any source point $s \in \mathbb{R}^d$ and its charge $q_s$, update the data structure by adding this source point in $\log^{O(d)}(\|q\|_1/\varepsilon)$ time.*

- DELETE$(s \in \mathbb{R}^d)$ *(Algorithm 5) For any source point $s \in \mathbb{R}^d$ and its charge $q_s$, update the data structure by deleting this source point in $\log^{O(d)}(\|q\|_1/\varepsilon)$ time.*

- QUERY$(q \in \mathbb{R}^N)$ *(Algorithm 3) Output $\widetilde{\mathsf{K}q} \in \mathbb{R}^N$ such that $\|\widetilde{\mathsf{K}q} - \mathsf{K}q\|_\infty \leq \varepsilon$, where $\mathsf{K} \in \mathbb{R}^{N \times N}$ is defined by $\mathsf{K}_{i,j} = \mathsf{K}(s_i, s_j)$ in $N \log^{O(d)}(\|q\|_1/\varepsilon)$ time.*

**Remark B.6.** *The* QUERY *time can be further reduced when the change of the charge vector $q$ is sparsely changed between two consecutive queries. More specifically, let $\Delta := \|q^{\text{new}} - q\|_0$ be the number of changed coordinates of $q$. Then,* QUERY *can be done in $\widetilde{O}_d(\Delta)$ time.*

## C ALGORITHMS

---

**Algorithm 2** This algorithm are the init part of Theorem B.5.

---

1: **data structure** DYNAMICFGT $\qquad\qquad\qquad\qquad\qquad\qquad\qquad$ ▷ Theorem B.5
2: **members**
3: $A_\alpha(\mathcal{B}_k), k \in [N(B)], \alpha \leq p$
4: $C_\beta(\mathcal{C}_k), k \in [N(C)], \beta \leq p$
5: $t_{\mathcal{C}_k}, k \in [N(C)]$
6: $s_{\mathcal{B}_k}, k \in [N(B)]$
7: **end members**
8:
9: **procedure** INIT$(\{s_j \in \mathbb{R}^d, j \in [N]\}, \{q_j \in \mathbb{R}, j \in [N]\})$
10: $\quad p \leftarrow \log(\|q\|_1/\varepsilon)$
11: $\quad$ Assign $N$ sources into $N(B)$ boxes $\mathcal{B}_1, \ldots, \mathcal{B}_{N(B)}$ of length $r\sqrt{\delta}$
12: $\quad$ Divide space into $N(C)$ boxes $\mathcal{C}_1, \ldots, \mathcal{C}_{N(C)}$ of length $r\sqrt{\delta}$
13: $\quad$ Set center $s_{\mathcal{B}_k}, k \in [N(B)]$ of source boxes $\mathcal{B}_1, \ldots, \mathcal{B}_{N(B)}$
14: $\quad$ Set centers $t_{\mathcal{C}_k}, k \in [N(C)]$ of target boxes $\mathcal{C}_1, \ldots, \mathcal{C}_{N(C)}$
15: $\quad$ **for** $k \in [N(B)]$ **do** $\qquad\qquad\qquad\qquad\qquad$ ▷ Source box $\mathcal{B}_k$ with center $s_{\mathcal{B}_k}$
16: $\quad\quad$ **for** $\alpha \leq p$ **do** $\qquad\qquad\qquad\qquad\qquad$ ▷ we say $\alpha \leq p$ if $\alpha_i \leq p, \forall i \in [d]$
17: $\quad\quad\quad$ Compute

$$A_\alpha(\mathcal{B}_k) \leftarrow \frac{(-1)^{\|\alpha\|_1}}{\alpha!} \sum_{s_j \in \mathcal{B}_k} q_j \left( \frac{s_j - s_{\mathcal{B}_k}}{\sqrt{\delta}} \right)^\alpha$$

$\qquad\qquad\qquad\qquad\qquad\qquad\qquad\qquad\qquad\qquad$ ▷ Takes $p^d N$ time in total

18: $\quad\quad$ **end for**
19: $\quad$ **end for**
20: $\quad$ **for** $k \in [N(C)]$ **do** $\qquad\qquad\qquad\qquad\qquad$ ▷ Target box $\mathcal{C}_k$ with center $t_{\mathcal{C}_k}$
21: $\quad\quad$ Find $(2k+1)^d$ nearest source boxes to $\mathcal{C}_k$, denote by $\mathsf{nb}(\mathcal{C}_k)$ $\quad$ ▷ $k \leq \log(\|q\|_1/\varepsilon)$
22: $\quad\quad$ **for** $\beta \leq p$ **do**
23: $\quad\quad\quad$ Compute

$$C_\beta(\mathcal{C}_k) \leftarrow \frac{(-1)^{\|\beta\|_1}}{\beta!} \sum_{\mathcal{B}_l \in \mathsf{nb}(\mathcal{C}_k)} \sum_{\alpha \leq p} A_\alpha(\mathcal{B}_l) \cdot H_{\alpha+\beta} \left( \frac{s_{\mathcal{B}_l} - t_{\mathcal{C}_k}}{\sqrt{\delta}} \right)$$

$\qquad\qquad\qquad\qquad\qquad\qquad\qquad$ ▷ Takes $N(C) \cdot (2k+1)^d d p^{d+1}$ time in total
24: $\qquad\qquad\qquad\qquad\qquad\qquad\qquad\qquad\qquad$ ▷ $N(C) \leq \min\{(r\sqrt{2\delta})^{-d/2}, M\}$
25: $\quad\quad$ **end for**
26: $\quad$ **end for**
27: **end procedure**
28: **end data structure**

---

---

**Algorithm 3** This algorithm is the query part of Theorem B.5.

---

1: **data structure** DYNAMICFGT
2: **procedure** KDE-QUERY($t \in \mathbb{R}^d$)
3:      Find the box $t \in \mathcal{C}_k$
4:      Compute                                     $\triangleright$ Takes $p^d$ time in total

$$G_p(t) \leftarrow \sum_{\beta \leq p} C_\beta(\mathcal{C}_k) \cdot \left( \frac{t - t_{\mathcal{C}_k}}{\sqrt{\delta}} \right)^\beta$$

5:      **return** $G_p(t)$
6: **end procedure**
7: **procedure** QUERY($q \in \mathbb{R}^N$)
8:      INIT($\{s_j, j \in [N]\}, q$)                         $\triangleright$ Takes $\widetilde{O}(N)$ time
9:      **for** $j \in [N]$ **do**
10:          $u_j \leftarrow$ LOCAL-QUERY($s_j$)            $\triangleright \|u - \mathsf{K}q\|_\infty \leq \varepsilon$
11:      **end for**
12:      **return** $u$
13: **end procedure**
14: **end data structure**

---

**Algorithm 4** This algorithm is the update part of Theorem B.5.

1: **data structure** DYNAMICFGT                                        ▷ Theorem B.5
2: **members**                        ▷ This is exact same as the members in Algorithm 2.
3:      $A_\alpha(\mathcal{B}_k), k \in [N(B)], \alpha \le p$
4:      $C_\beta(\mathcal{C}_k), k \in [N(C)], \beta \le p$
5:      $t_{\mathcal{C}_k}, k \in [N(C)]$
6:      $s_{\mathcal{B}_k}, k \in [N(B)]$
7: **end members**
8:
9: **procedure** INSERT($s \in \mathbb{R}^d, q \in \mathbb{R}$)
10:      Find the box $s \in \mathcal{B}_k$
11:      **for** $\alpha \le p$ **do**                             ▷ we say $\alpha \le p$ if $\alpha_i \le p, \forall i \in [d]$
12:          Compute

$$A_\alpha^{\text{new}}(\mathcal{B}_k) \leftarrow A_\alpha(\mathcal{B}_k) + \frac{(-1)^{\|\alpha\|_1} q}{\alpha!} \left(\frac{s - s_{\mathcal{B}_k}}{\sqrt{\delta}}\right)^\alpha$$

                                                          ▷ Takes $p^d$ time
13:      **end for**
14:      Find $(2k+1)^d$ nearest target boxes to $\mathcal{B}_k$, denote by nb($\mathcal{B}_k$)     ▷ $k \le \log(\|q\|_1/\varepsilon)$
15:      **for** $\mathcal{C}_l \in$ nb($\mathcal{B}_k$) **do**
16:          **for** $\beta \le p$ **do**
17:              Compute

$$C_\beta^{\text{new}}(\mathcal{C}_l) \leftarrow C_\beta(\mathcal{C}_l) + \frac{(-1)^{\|\beta\|_1}}{\beta!} \sum_{\alpha \le p} (A_\alpha^{\text{new}}(\mathcal{B}_k) - A_\alpha(\mathcal{B}_k)) \cdot H_{\alpha+\beta}\left(\frac{s_{\mathcal{B}_k} - t_{\mathcal{C}_l}}{\sqrt{\delta}}\right)$$

                                                     ▷ Takes $(2k+1)^d p^d$ time
18:          **end for**
19:      **end for**
20:      **for** $\alpha \le p$ **do**
21:          $A_\alpha(\mathcal{B}_k) \leftarrow A_\alpha^{\text{new}}(\mathcal{B}_k)$                                 ▷ Takes $p^d$ time
22:      **end for**
23:      **for** $\mathcal{C}_l \in$ nb($\mathcal{B}_k$) **do**
24:          **for** $\beta \le p$ **do**
25:              $C_\beta(\mathcal{C}_l) \leftarrow C_\beta^{\text{new}}(\mathcal{C}_l)$                       ▷ Takes $(2k+1)^d p^d$ time
26:          **end for**
27:      **end for**
28: **end procedure**
29: **end data structure**

---

**Algorithm 5** This algorithm is another update part of Theorem B.5.

---

1: **data structure** DYNAMICFGT
2: **members**
3:     $A_\alpha(\mathcal{B}_k), k \in [N(B)], \alpha \leq p$
4:     $C_\beta(\mathcal{C}_k), k \in [N(C)], \beta \leq p$
5:     $t_{\mathcal{C}_k}, k \in [N(C)]$
6:     $s_{\mathcal{B}_k}, k \in [N(B)]$
7:     $\delta \in \mathbb{R}$
8: **end members**
9:
10: **procedure** DELETE($s \in \mathbb{R}^d, q \in \mathbb{R}$)
11:     Find the box $s \in \mathcal{B}_k$
12:     **for** $\alpha \leq p$ **do**                                        ▷ we say $\alpha \leq p$ if $\alpha_i \leq p, \forall i \in [d]$
13:         Compute

$$A_\alpha^{\text{new}}(\mathcal{B}_k) \leftarrow A_\alpha(\mathcal{B}_k) - \frac{(-1)^{\|\alpha\|_1} q}{\alpha!} \left( \frac{s - s_{\mathcal{B}_k}}{\sqrt{\delta}} \right)^\alpha$$

▷ Takes $p^d$ time

14:     **end for**
15:     Find $(2k+1)^d$ nearest target boxes to $\mathcal{B}_k$, denote by $\mathsf{nb}(\mathcal{B}_k)$          ▷ $k \leq \log(\|q\|_1/\varepsilon)$
16:     **for** $\mathcal{C}_l \in \mathsf{nb}(\mathcal{B}_k)$ **do**
17:         **for** $\beta \leq p$ **do**
18:             Compute

$$C_\beta^{\text{new}}(\mathcal{C}_l) \leftarrow C_\beta(\mathcal{C}_l) + \frac{(-1)^{\|\beta\|_1}}{\beta!} \sum_{\alpha \leq p} (A_\alpha^{\text{new}}(\mathcal{B}_k) - A_\alpha(\mathcal{B}_k)) \cdot H_{\alpha+\beta} \left( \frac{s_{\mathcal{B}_k} - t_{\mathcal{C}_l}}{\sqrt{\delta}} \right)$$

▷ Takes $(2k+1)^d p^d$ time

19:         **end for**
20:     **end for**
21:     **for** $\alpha \leq p$ **do**
22:         $A_\alpha(\mathcal{B}_k) \leftarrow A_\alpha^{\text{new}}(\mathcal{B}_k)$                        ▷ Takes $p^d$ time
23:     **end for**
24:     **for** $\mathcal{C}_l \in \mathsf{nb}(\mathcal{B}_k)$ **do**
25:         **for** $\beta \leq p$ **do**
26:             $C_\beta(\mathcal{C}_l) \leftarrow C_\beta^{\text{new}}(\mathcal{C}_l)$                ▷ Takes $(2k+1)^d p^d$ time
27:         **end for**
28:     **end for**
29: **end procedure**
30: **end data structure**

---

# D ANALYSIS

*Proof of Theorem B.5.* **Correctness of KDE-QUERY.** Algorithm 2 accumulates all sources into truncated Hermite expansions and transforms all Hermite expansions into Taylor expansions via Lemma E.5, thus it can approximate the function $G(t)$ by

$$G(t) = \sum_{\mathcal{B}} \sum_{j \in \mathcal{B}} q_j \cdot e^{-\|t - s_j\|_2^2 / \delta}$$

$$= \sum_{\beta \leq p} C_\beta \left( \frac{t - t_{\mathcal{C}}}{\sqrt{\delta}} \right)^\beta + \mathrm{Err}_T(p) + \mathrm{Err}_H(p)$$

where $|\mathrm{Err}_H(p)| + |\mathrm{Err}_T(p)| \leq Q \cdot \varepsilon$ by $p = \log(\|q\|_1 / \varepsilon)$,

$$C_\beta = \frac{(-1)^{\|\beta\|_1}}{\beta!} \sum_{\mathcal{B}} \sum_{\alpha \leq p} A_\alpha(\mathcal{B}) H_{\alpha + \beta} \left( \frac{s_{\mathcal{B}} - t_{\mathcal{C}}}{\sqrt{\delta}} \right)$$

and the coefficients $A_\alpha(\mathcal{B})$ are defined as Eq. (3).

**Running time of KDE-QUERY.** In line 17, it takes $O(p^d N)$ time to compute all the Hermite expansions, i.e., to compute the coefficients $A_\alpha(\mathcal{B})$ for all $\alpha \leq p$ and all sources boxes $\mathcal{B}$.

Making use of the large product in the definition of $H_{\alpha + \beta}$, we see that the time to compute the $p^d$ coefficients of $C_\beta$ is only $O(dp^{d+1})$ for each box $\mathcal{B}$ in the range. Thus, we know for each target box $\mathcal{C}$, the running time is $O((2k + 1)^d dp^{d+1})$, thus the total time in line 23 is

$$O(N(C) \cdot (2k + 1)^d dp^{d+1}).$$

Finally we need to evaluate the appropriate Taylor series for each target $t_i$, which can be done in $O(p^d M)$ time in line 4. Putting it all together, Algorithm 2 takes time

$$O((2k + 1)^d dp^{d+1} N(C)) + O(p^d N) + O(p^d M)$$
$$= O\left( (M + N) \log^{O(d)}(\|q\|_1 / \varepsilon) \right).$$

**Correctness of UPDATE.** Algorithm 4 and Algorithm 5 maintains $C_\beta$ as follows,

$$C_\beta = \frac{(-1)^{\|\beta\|_1}}{\beta!} \sum_{\mathcal{B}} \sum_{\alpha \leq p} A_\alpha(\mathcal{B}) H_{\alpha + \beta} \left( \frac{s_{\mathcal{B}} - t_{\mathcal{C}}}{\sqrt{\delta}} \right)$$

where $A_\alpha(\mathcal{B})$ is given by

$$A_\alpha(\mathcal{B}) = \frac{(-1)^{\|\alpha\|_1}}{\alpha!} \sum_{j \in \mathcal{B}} q_j \cdot \left( \frac{s_j - s_{\mathcal{B}}}{\sqrt{\delta}} \right)^\alpha.$$

Therefore, the correctness follows similarly from Algorithm 2.

**Running time of UPDATE.** In line 12, it takes $O(p^d)$ time to update all the Hermite expansions, i.e. to update the coefficients $A_\alpha(\mathcal{B})$ for all $\alpha \leq p$ and all sources boxes $\mathcal{B}$.

Making use of the large product in the definition of $H_{\alpha + \beta}$, we see that the time to compute the $p^d$ coefficients of $C_\beta$ is only $O(dp^{d+1})$ for each box $\mathcal{C}_l \in \mathsf{nb}(\mathcal{B}_k)$. Thus, thus the total time in line 17 is

$$O((2k + 1)^d dp^{d+1}).$$

**Correctness of QUERY.** To compute $\mathsf{K}q$ for a given $q \in \mathbb{R}^d$, notice that for any $i \in [N]$,

$$(\mathsf{K}q)_i = \sum_{j=1}^N q_j \cdot e^{-\|s_i - s_j\|_2^2 / \delta}$$

$$= G(s_i).$$

Hence, this problem reduces to $N$ KDE-QUERY() calls. And the additive error guarantee for $G(t)$ immediately gives the $\ell_\infty$-error guarantee for $\mathsf{K}q$.

**Running time of QUERY.** We first build the data structure with the charge vector $q$ given in the query, which takes $\widetilde{O}_d(N)$ time. Then, we perform $N$ KDE-Query, each takes $\widetilde{O}_d(1)$. Hence, the total running time is $\widetilde{O}_d(N)$.

We note that when the charge vector $q$ is slowly changing, i.e., $\Delta := \|q^{\text{new}} - q\|_0 \leq o(N)$, we can UPDATE the source vectors whose charges are changed. Since each INSERT or DELETE takes $\widetilde{O}_d(1)$ time, it will take $\widetilde{O}_d(\Delta)$ time to update the data structure.

Then, consider computing $\mathsf{K}q^{\text{new}}$ in this setting. We note that each source box can only affect $\widetilde{O}_d(1)$ other target boxes, where the target vectors are just the source vectors in this setting. Hence, there are at most $\widetilde{O}_d(\Delta)$ boxes whose $C_\beta$ is changed. Let $\mathcal{S}$ denote the indices of source vectors in these boxes. Since

$$G(s_i) = \sum_{\beta \leq p} C_\beta(\mathcal{B}_k) \cdot \left( \frac{s_i - s_{\mathcal{B}_k}}{\sqrt{\delta}} \right)^\beta,$$

we get that there are at most $\widetilde{O}_d(\Delta)$ coordinates of $\mathsf{K}q^{\text{new}}$ that are significantly changed from $\mathsf{K}q$, and we only need to re-compute $G(s_i)$ for $i \in \mathcal{S}$. If we assume that the source vectors are well-separated, i.e., $|\mathcal{S}| = O(\delta)$, the total computational cost is $\widetilde{O}_d(\Delta)$.

Therefore, when the change of the charge vector $q$ is sparse, $\mathsf{K}q$ can be computed in sublinear time. $\qquad\square$

## E    ERROR ESTIMATION

This section provides several technical lemma that are used in Appendix D. We first give a definition.

**Definition E.1** (Hermite expansion and coefficients). *Let $\mathcal{B}$ denote a box with center $s_\mathcal{B} \in \mathbb{R}^d$ and side length $r\sqrt{2\delta}$ with $r < 1$. If source $s_j$ is in box $\mathcal{B}$, we will simply denote as $j \in \mathcal{B}$. Then the Gaussian evaluation from the sources in box $\mathcal{B}$ is,*

$$G(t) = \sum_{j \in \mathcal{B}} q_j \cdot e^{-\|t - s_j\|_2^2/\delta}.$$

*The Hermite expansion of $G(t)$ is*

$$G(t) = \sum_{\alpha \geq 0} A_\alpha \cdot H_\alpha\left( \frac{t - s_\mathcal{B}}{\sqrt{\delta}} \right), \tag{6}$$

*where the coefficients $A_\alpha$ are defined by*

$$A_\alpha = \frac{1}{\alpha!} \sum_{j \in \mathcal{B}} q_j \cdot \left( \frac{s_j - s_\mathcal{B}}{\sqrt{\delta}} \right)^\alpha \tag{7}$$

The rest of this section will present a batch of Lemmas that bound the error of the function truncated at certain degree of Taylor and Hermite expansion.

We first upper bound the truncation error of Hermite expansion.

**Lemma E.2** (Truncated Hermite expansion). *Let $p$ denote an integer, let $\text{Err}_H(p)$ denote the error after truncating the series $G(t)$ (as defined in Eq. (6)) after $p^d$ terms, i.e.,*

$$\text{Err}_H(p) = \sum_{\alpha \geq p} A_\alpha \cdot H_\alpha\left( \frac{t - s_\mathcal{B}}{\sqrt{\delta}} \right). \tag{8}$$

*Then we have*

$$|\text{Err}_H(p)| \leq \frac{\sum_{j \in \mathcal{B}} |q_j|}{(1-r)^d} \sum_{k=0}^{d-1} \binom{d}{k} (1 - r^p)^k \left( \frac{r^p}{\sqrt{p!}} \right)^{d-k}$$

*where $r \leq \frac{1}{2}$.*

*Proof.* Using Eq. (4) to expand each Gaussian (see Definition E.1) in the

$$G(t) = \sum_{j \in \mathcal{B}} q_j \cdot e^{-\|t - s_j\|_2^2 / \delta}$$

into a Hermite series about $s_{\mathcal{B}}$:

$$\sum_{j \in \mathcal{B}} q_j \sum_{\alpha \geq 0} \frac{1}{\alpha!} \cdot \left( \frac{s_j - s_{\mathcal{B}}}{\sqrt{\delta}} \right)^{\alpha} \cdot H_{\alpha} \left( \frac{t - s_{\mathcal{B}}}{\sqrt{\delta}} \right)$$

and swap the summation over $\alpha$ and $j$ to obtain the desired form:

$$\sum_{\alpha \geq 0} \left( \frac{1}{\alpha!} \sum_{j \in \mathcal{B}} q_j \cdot \left( \frac{s_j - s_{\mathcal{B}}}{\sqrt{\delta}} \right)^{\alpha} \right) H_{\alpha} \left( \frac{t - s_{\mathcal{B}}}{\sqrt{\delta}} \right) = \sum_{\alpha \geq 0} A_{\alpha} H_{\alpha} \left( \frac{t - s_{\mathcal{B}}}{\sqrt{\delta}} \right).$$

Here, the truncation error bound is due to Lemma A.8 and the standard equation for the tail of a geometric series.

To formally bound the truncation error, we first rewrite the Hermit expansion as follows

$$e^{-\frac{\|t - s_j\|_2^2}{\delta}} = \prod_{i=1}^{d} \left( \sum_{n_i=1}^{p-1} \frac{1}{n_i!} \left( \frac{(s_j)_i - (s_{\mathcal{B}})_i}{\sqrt{\delta}} \right)^{n_i} h_{n_i} \left( \frac{t_i - (s_{\mathcal{B}})_i}{\sqrt{\delta}} \right) \right.$$

$$\left. + \sum_{n_i=p}^{\infty} \frac{1}{n_i!} \left( \frac{(s_j)_i - (s_{\mathcal{B}})_i}{\sqrt{\delta}} \right)^{n_i} h_{n_i} \left( \frac{t_i - (s_{\mathcal{B}})_i}{\sqrt{\delta}} \right) \right) \tag{9}$$

Notice from Cramer's inequality (Lemma A.5),

$$h_{n_i} \left( \frac{t_i - (s_{\mathcal{B}})_i}{\sqrt{\delta}} \right) \leq \sqrt{n!} \cdot 2^{n/2} \cdot e^{-(t_i - (s_{\mathcal{B}})_i)^2/2}.$$

Therefore we can use properties of the geometric series (notice $\frac{(s_j)_i - (s_{\mathcal{B}})_i}{\sqrt{\delta}} \leq r/\sqrt{2}$) to bound each term in the product as follows

$$\sum_{n_i=1}^{p-1} \frac{1}{n_i!} \left( \frac{(s_j)_i - (s_{\mathcal{B}})_i}{\sqrt{\delta}} \right)^{n_i} h_{n_i} \left( \frac{t_i - (s_{\mathcal{B}})_i}{\sqrt{\delta}} \right) \leq \frac{1 - r^p}{1 - r}, \tag{10}$$

and

$$\sum_{n_i=p}^{\infty} \frac{1}{n_i!} \left( \frac{(s_j)_i - (s_{\mathcal{B}})_i}{\sqrt{\delta}} \right)^{n_i} h_{n_i} \left( \frac{t_i - (s_{\mathcal{B}})_i}{\sqrt{\delta}} \right) \leq \frac{1}{\sqrt{p!}} \cdot \frac{r^p}{1 - r}. \tag{11}$$

Now we come back to bound Eq. (8) as follows

$$\mathrm{Err}_H(p) = \sum_{j \in \mathcal{B}} q_j \sum_{\alpha \geq p} \frac{1}{\alpha!} \cdot \left( \frac{s_j - s_{\mathcal{B}}}{\sqrt{\delta}} \right)^{\alpha} \cdot H_{\alpha} \left( \frac{t - s_{\mathcal{B}}}{\sqrt{\delta}} \right)$$

$$\leq \left( \sum_{j \in \mathcal{B}} |q_j| \right) \left( e^{-\frac{\|t - s_j\|_2^2}{\delta}} - \prod_{j=1}^{d} \left( \sum_{n_i=1}^{p-1} \frac{1}{n_i!} \left( \frac{(s_j)_i - (s_{\mathcal{B}})_i}{\sqrt{\delta}} \right)^{n_i} h_{n_i} \left( \frac{t_i - (s_{\mathcal{B}})_i}{\sqrt{\delta}} \right) \right) \right)$$

$$\leq \frac{\sum_{j \in \mathcal{B}} |q_j|}{(1 - r)^d} \sum_{k=0}^{d-1} \binom{d}{k} (1 - r^p)^k \left( \frac{r^p}{\sqrt{p!}} \right)^{d-k}$$

where the first step comes from definition, the second step comes from Eq. (9) and the last step comes from Eq. (10) and Eq. (11) and binomial expansion.

$\square$

**Remark E.3.** *By Stirling's formula, it is easy to see that when we take $p = \log(\|q\|_1 / \varepsilon)$, this error will be bounded by $\|q\|_1 \cdot \varepsilon$.*

The Lemma E.4 shows how to convert a Hermite expansion at location $s_\mathcal{B}$ into a Taylor expansion at location $t_\mathcal{C}$. Intuitively, the Taylor series converges rapidly in the box (that has side length $r\sqrt{2\delta}$ center around $t_\mathcal{C}$, where $r \in (0,1)$).

**Lemma E.4** (Hermite expansion with truncated Taylor expansion). *Suppose the Hermite expansion of $G(t)$ is given by Eq. (6), i.e.,*

$$G(t) = \sum_{\alpha \geq 0} A_\alpha \cdot H_\alpha \left( \frac{t - s_\mathcal{B}}{\sqrt{\delta}} \right). \tag{12}$$

*Then, the Taylor expansion of $G(t)$ at an arbitrary point $t_0$ can be written as:*

$$G(t) = \sum_{\beta \geq 0} B_\beta \left( \frac{t - t_0}{\sqrt{\delta}} \right)^\beta. \tag{13}$$

*where the coefficients $B_\beta$ are defined as*

$$B_\beta = \frac{(-1)^{\|\beta\|_1}}{\beta!} \sum_{\alpha \geq 0} (-1)^{\|\alpha\|_1} A_\alpha \cdot H_{\alpha+\beta} \left( \frac{s_\mathcal{B} - t_0}{\sqrt{\delta}} \right). \tag{14}$$

*Let $\mathrm{Err}_T(p)$ denote the error by truncating the Taylor expansion after $p^d$ terms, in the box $\mathcal{C}$ (that has center at $t_\mathcal{C}$ and side length $r\sqrt{2\delta}$), i.e.,*

$$\mathrm{Err}_T(p) = \sum_{\beta \geq p} B_\beta \left( \frac{t - t_\mathcal{C}}{\sqrt{\delta}} \right)^\beta$$

*Then*

$$|\mathrm{Err}_T(p)| \leq \frac{\sum_{j \in \mathcal{B}} |q_j|}{(1-r)^d} \sum_{k=0}^{d-1} \binom{d}{k} (1 - r^p)^k \left( \frac{r^p}{\sqrt{p!}} \right)^{d-k}$$

*where $r \leq 1/2$.*

*Proof.* Each Hermite function in Eq. (12) can be expanded into a Taylor series by means of Eq. (5). The expansion in Eq. (13) is due to swapping the order of summation.

Next, we will bound the truncation error. Using Eq. (7) for $A_\alpha$, we can rewrite $B_\beta$:

$$B_\beta = \frac{(-1)^{\|\beta\|_1}}{\beta!} \sum_{\alpha \geq 0} (-1)^{\|\alpha\|_1} A_\alpha H_{\alpha+\beta} \left( \frac{s_\mathcal{B} - t_\mathcal{C}}{\sqrt{\delta}} \right)$$

$$= \frac{(-1)^{\|\beta\|_1}}{\beta!} \sum_{\alpha \geq 0} \left( \frac{(-1)^{\|\alpha\|_1}}{\alpha!} \sum_{j \in \mathcal{B}} q_j \left( \frac{s_j - s_\mathcal{B}}{\sqrt{\delta}} \right)^\alpha \right) H_{\alpha+\beta} \left( \frac{s_\mathcal{B} - t_\mathcal{C}}{\sqrt{\delta}} \right)$$

$$= \frac{(-1)^{\|\beta\|_1}}{\beta!} \sum_{j \in \mathcal{B}} q_j \sum_{\alpha \geq 0} \frac{(-1)^{\|\alpha\|_1}}{\alpha!} \left( \frac{s_j - s_\mathcal{B}}{\sqrt{\delta}} \right)^\alpha \cdot H_{\alpha+\beta} \left( \frac{s_\mathcal{B} - t_\mathcal{C}}{\sqrt{\delta}} \right)$$

By Eq. (5), the inner sum is the Taylor expansion of $H_\beta((s_j - t_\mathcal{C})/\sqrt{\delta})$. Thus

$$B_\beta = \frac{(-1)^{\|\beta\|_1}}{\beta!} \sum_{j \in \mathcal{B}} q_j \cdot H_\beta \left( \frac{s_j - t_\mathcal{C}}{\sqrt{\delta}} \right)$$

and Cramer's inequality implies

$$|B_\beta| \leq \frac{1}{\beta!} K \cdot Q_B 2^{\|\beta\|_1/2} \sqrt{\beta!} = K Q_B \frac{2^{\|\beta\|_1/2}}{\sqrt{\beta!}}.$$

To formally bound the truncation error, we have

$$\mathrm{Err}_T(p) = \sum_{\beta \geq p} B_\beta \left( \frac{t - t_\mathcal{C}}{\sqrt{\delta}} \right)^\beta$$

$$\leq KQ_B \left( \prod_{i=1}^{d} \left( \sum_{n_i=0}^{\infty} \frac{1}{\sqrt{n_i!}} 2^{n_i/2} \left( \frac{t-t_\mathcal{C}}{\delta} \right)^{n_i} \right) - \prod_{i=1}^{d} \left( \sum_{n_i=0}^{p-1} \frac{1}{\sqrt{n_i!}} 2^{n_i/2} \left( \frac{t-t_\mathcal{C}}{\delta} \right)^{n_i} \right) \right)$$

$$\leq \frac{\sum_{j \in \mathcal{B}} |q_j|}{(1-r)^d} \sum_{k=0}^{d-1} \binom{d}{k} (1-r^p)^k \left( \frac{r^p}{\sqrt{p!}} \right)^{d-k}$$

where the second step uses $|B_\beta| \leq KQ_B \frac{2^{\|\beta\|_1/2}}{\sqrt{\beta!}}$ and the rest are similar to those in Lemma E.2. $\qquad\square$

For designing our algorithm, we would like to make a variant of Lemma E.4 that combines the truncations of Hermite expansion and Taylor expansion. More specifically, we first truncate the Taylor expansion of $G_p(t)$, and then truncate the Hermite expansion in Eq. (14) for the coefficients.

**Lemma E.5** (Truncated Hermite expansion with truncated Taylor expansion). *Let $G(t)$ be defined as Def E.1. For an integer $p$, let $G_p(t)$ denote the Hermite expansion of $G(t)$ truncated at $p$, i.e.,*

$$G_p(t) = \sum_{\alpha \leq p} A_\alpha H_\alpha \left( \frac{t-s_\mathcal{B}}{\sqrt{\delta}} \right).$$

*The Taylor expansion of function $G_p(t)$ at an arbitrary point $t_0$ can be written as:*

$$G_p(t) = \sum_{\beta \geq 0} C_\beta \cdot \left( \frac{t-t_0}{\sqrt{\delta}} \right)^\beta,$$

*where the coefficients $C_\beta$ are defined as*

$$C_\beta = \frac{(-1)^{\|\beta\|_1}}{\beta!} \sum_{\alpha \leq p} (-1)^{\|\alpha\|_1} A_\alpha \cdot H_{\alpha+\beta} \left( \frac{s_\mathcal{B}-t_\mathcal{C}}{\sqrt{\delta}} \right). \tag{15}$$

*Let $\mathrm{Err}_T(p)$ denote the error in truncating the Taylor series after $p^d$ terms, in the box $\mathcal{C}$ (that has center $t_\mathcal{C}$ and side length $r\sqrt{2\delta}$), i.e.,*

$$\mathrm{Err}_T(p) = \sum_{\beta \geq p} C_\beta \left( \frac{t-t_\mathcal{C}}{\sqrt{\delta}} \right)^\beta.$$

*Then, we have*

$$|\mathrm{Err}_T(p)| \leq \frac{2 \sum_{j \in \mathcal{B}} |q_j|}{(1-r)^d} \sum_{k=0}^{d-1} \binom{d}{k} (1-r^p)^k \left( \frac{r^p}{\sqrt{p!}} \right)^{d-k}$$

*where $r \leq 1/2$.*

*Proof.* We can write $C_\beta$ in the following way:

$$C_\beta = \frac{(-1)^{\|\beta\|_1}}{\beta!} \sum_{j \in \mathcal{B}} q_j \sum_{\alpha \leq p} \frac{(-1)^{\|\alpha\|_1}}{\alpha!} \left( \frac{s_j - s_\mathcal{B}}{\sqrt{\delta}} \right)^\alpha \cdot H_{\alpha+\beta} \left( \frac{s_\mathcal{B}-t_\mathcal{C}}{\sqrt{\delta}} \right)$$

$$= \frac{(-1)^{\|\beta\|_1}}{\beta!} \sum_{j \in \mathcal{B}} q_j \left( \sum_{\alpha \geq 0} - \sum_{\alpha > p} \right) \frac{(-1)^{\|\alpha\|_1}}{\alpha!} \left( \frac{s_j - s_\mathcal{B}}{\sqrt{\delta}} \right)^\alpha \cdot H_{\alpha+\beta} \left( \frac{s_\mathcal{B}-t_\mathcal{C}}{\sqrt{\delta}} \right)$$

$$= B_\beta - \frac{(-1)^{\|\beta\|_1}}{\beta!} \sum_{j \in \mathcal{B}} q_j \sum_{\alpha > p} \frac{(-1)^{\|\alpha\|_1}}{\alpha!} \left( \frac{s_j - s_\mathcal{B}}{\sqrt{\delta}} \right)^\alpha \cdot H_{\alpha+\beta} \left( \frac{s_\mathcal{B}-t_\mathcal{C}}{\sqrt{\delta}} \right)$$

$$= B_\beta + (C_\beta - B_\beta)$$

Next, we have

$$|\mathrm{Err}_T(p)| \leq \left| \sum_{\beta \geq p} B_\beta \left( \frac{t-t_\mathcal{C}}{\sqrt{\delta}} \right)^\beta \right| + \left| \sum_{\beta \geq p} (C_\beta - B_\beta) \cdot \left( \frac{t-t_\mathcal{C}}{\sqrt{\delta}} \right)^\beta \right| \tag{16}$$

Using Lemma E.4, we can upper bound the first term in the Eq. (16) by,

$$\left| \sum_{\beta \geq p} B_\beta \left( \frac{t - t_C}{\sqrt{\delta}} \right)^\beta \right| \leq \frac{\sum_{j \in \mathcal{B}} |q_j|}{(1 - r)^d} \sum_{k=0}^{d-1} \binom{d}{k} (1 - r^p)^k \left( \frac{r^p}{\sqrt{p!}} \right)^{d-k}.$$

Since we can similarly bound $C_\beta - B_\beta$ as follows

$$|C_\beta - B_\beta| \leq \frac{1}{\beta!} K \cdot Q_B 2^{\|\beta\|_1/2} \sqrt{\beta!} \leq K Q_B \frac{2^{\|\beta\|_1/2}}{\sqrt{\beta!}},$$

we have the same bound for the second term

$$\left| \sum_{\beta \geq p} (C_\beta - B_\beta) \left( \frac{t - t_C}{\sqrt{\delta}} \right)^\beta \right| \leq \frac{\sum_{j \in \mathcal{B}} |q_j|}{(1 - r)^d} \sum_{k=0}^{d-1} \binom{d}{k} (1 - r^p)^k \left( \frac{r^p}{\sqrt{p!}} \right)^{d-k}.$$

$\square$

The proof of the following Lemma is almost identical, but it directly bounds the truncation error of Taylor expansion of the Gaussian kernel. We omit the proof here.

**Lemma E.6** (Truncated Taylor expansion). *Let $G_{s_j}(t) : \mathbb{R}^d \to \mathbb{R}$ be defined as*

$$G_{s_j}(t) = q_j \cdot e^{-\|t - s_j\|_2^2/\delta}.$$

*The Taylor expansion of $G_{s_j}(t)$ at $t_C \in \mathbb{R}^d$ is:*

$$G_{s_j}(t) = \sum_{\beta \geq 0} \mathcal{B}_\beta \left( \frac{t - t_C}{\sqrt{\delta}} \right)^\beta,$$

*where the coefficients $B_\beta$ is defined as*

$$B_\beta = q_j \cdot \frac{(-1)^{\|\beta\|_1}}{\beta!} \cdot H_\beta \left( \frac{s_j - t_C}{\sqrt{\delta}} \right)$$

*and the absolute value of the error (truncation after $p^d$ terms) can be upper bounded as*

$$|\mathrm{Err}_T(p)| \leq \frac{\sum_{j \in \mathcal{B}} |q_j|}{(1 - r)^d} \sum_{k=0}^{d-1} \binom{d}{k} (1 - r^p)^k \left( \frac{r^p}{\sqrt{p!}} \right)^{d-k}$$

*where $r \leq 1/2$.*

## F  LOW DIMENSION SUBSPACE FGT

In this section, we consider FGT for data in a lower dimensional subspace of $\mathbb{R}^d$. The problem is formally defined below:

**Problem F.1** (Dynamic FGT on a low dimensional set). *Let $W$ be a subspace of $\mathbb{R}^d$ with dimension $\dim(S) = w \ll d$. Given $N$ source points $s_1, \ldots, s_N \in W$ with charges $q_1, \ldots, q_N$, and $M$ target points $t_1, \ldots, t_M \in W$, find a dynamic data structure that supports the following operations:*

- INSERT/DELETE$(s_i, q_i)$ *Insert or Delete a source point $s_i \in \mathbb{R}^d$ along with its "charge" $q_i \in \mathbb{R}$, in $\log^{O(w)}(\|q\|_1/\varepsilon)$ time.*

- DENSITY-ESTIMATION$(t \in \mathbb{R}^d)$ *For any point $t \in \mathbb{R}^d$, output the kernel density of $t$ with respect to the sources, i.e., output $\widetilde{G}$ such that $G(t) - \varepsilon \leq \widetilde{G} \leq G(t) + \varepsilon$ in $\log^{O(w)}(\|q\|_1/\varepsilon)$ time.*

- QUERY$(q \in \mathbb{R}^N)$ *Given an arbitrary query vector $q \in \mathbb{R}^N$, output $\widetilde{\mathsf{K}q}$ in $N \cdot \log^{O(w)}(\|q\|/\varepsilon)$ time.*

---

**Algorithm 6** Initialization of low-dim FGT.

---

1: **data structure** DYNAMICFGT
2: **members**
3: $A_\alpha(\mathcal{B}_i), i \in [N(B)], \alpha \leq p$
4: $C_\beta(\mathcal{C}_i), i \in [N(C)], \beta \leq p$
5: $t_{\mathcal{C}_i}, i \in [N(C)]$
6: $s_{\mathcal{B}_i}, i \in [N(B)]$
7: **end members**
8:
9: **procedure** INIT($\{s_j \in \mathbb{R}^d, j \in [N]\}, \{q_j \in \mathbb{R}, j \in [N]\}$)
10:     $p \leftarrow \log(\|q\|_1/\varepsilon)$
11:     Compute SVD: $(U_0, \Sigma, V_0) \leftarrow \text{SVD}\left((s_1, \ldots, s_N, t_1, \ldots, t_M)\right)$
12:     $\triangleright$

$U_0 \Sigma V_0^\top = (s_1, \ldots, s_N, t_1, \ldots, t_M), U_0 \in \mathbb{R}^{d \times d}, \Sigma \in \mathbb{R}^{d \times (N+M)}, V_0 \in \mathbb{R}^{(N+M) \times (N+M)}$
13:     Let $B \leftarrow U_0 \Sigma_{:,1:w} \in \mathbb{R}^{d \times w}$     $\triangleright$ $\Sigma_{:,1:w}$ denotes the first $w$ columns of $\Sigma$
14:     Compute the spectral decomposition $U \Lambda U^\top = B^\top B$, and let $\mathsf{P} \leftarrow \Lambda^{-1/2} U^{-1} B^\top \in \mathbb{R}^{w \times d}$
15:     **for** $i \in [N]$ and $j \in [M]$ **do**
16:         $x_i \leftarrow \mathsf{P} s_i, y_j \leftarrow \mathsf{P} t_j$
17:     **end for**
18:     Assign $x_1, \ldots, x_N$ into $N(B)$ boxes $\mathcal{B}_1, \ldots, \mathcal{B}_{N(B)}$ of length $r\sqrt{\delta}$
19:     Divide $\mathbb{R}^w$ into $N(C)$ boxes $\mathcal{C}_1, \ldots, \mathcal{C}_{N(C)}$ of length $r\sqrt{\delta}$
20:     Set center $x_{\mathcal{B}_i}, i \in [N(B)]$ of source boxes $\mathcal{B}_1, \ldots, \mathcal{B}_{N(B)}$
21:     Set centers $y_{\mathcal{C}_j}, j \in [N(C)]$ of target boxes $\mathcal{C}_1, \ldots, \mathcal{C}_{N(C)}$
22:     **for** $l \in [N(B)]$ **do**     $\triangleright$ Source box $\mathcal{B}_l$ with center $s_{\mathcal{B}_l}$
23:         **for** $\alpha \leq p$ **do**     $\triangleright$ we say $\alpha \leq p$ if $\alpha_i \leq p, \forall i \in [w]$
24:             Compute

$$A_\alpha(\mathcal{B}_l) \leftarrow \frac{(-1)^{\|\alpha\|_1}}{\alpha!} \sum_{x_j \in \mathcal{B}_l} q_j \left(\frac{x_j - x_{\mathcal{B}_l}}{\sqrt{\delta}}\right)^\alpha$$

$\triangleright$ Takes $p^w N$ time in total
25:         **end for**
26:     **end for**
27:     **for** $l \in [N(C)]$ **do**     $\triangleright$ Target box $\mathcal{C}_l$ with center $t_{\mathcal{C}_l}$
28:         Find $(2k+1)^w$ nearest source boxes to $\mathcal{C}_l$, denote by $\mathsf{nb}(\mathcal{C}_l)$     $\triangleright$ $k \leq \log(\|q\|_1/\varepsilon)$
29:         **for** $\beta \leq p$ **do**
30:             Compute

$$C_\beta(\mathcal{C}_l) \leftarrow \frac{(-1)^{\|\beta\|_1}}{\beta!} \sum_{\mathcal{B} \in \mathsf{nb}(\mathcal{C}_l)} \sum_{\alpha \leq p} A_\alpha(\mathcal{B}) \cdot H_{\alpha+\beta}\left(\frac{x_\mathcal{B} - y_{\mathcal{C}_l}}{\sqrt{\delta}}\right)$$

$\triangleright$ Takes $N(C) \cdot (2k+1)^w d p^{w+1}$ time in total
$\triangleright$ $N(C) \leq \min\{(r\sqrt{2\delta})^{-d/2}, M\}$
31:
32:         **end for**
33:     **end for**
34: **end procedure**
35: **end data structure**

---

We generalize our dynamic data structure to solve Problem F.1, which is stated in the following theorem. The computational cost of each update or query depends on the intrinsic dimension $w$ instead of $d$.

**Theorem F.2** (Low Rank Dynamic FGT Data Structure, formal version of Theorem 1.1)**.** *Let $W$ be a subspace of $\mathbb{R}^d$ with dimension $\dim(S) = w \ll d$. Given $N$ source points $s_1, \ldots, s_N \in W$ with charges $q_1, \ldots, q_N$, and $M$ target points $t_1, \ldots, t_M \in W$, a number $\delta > 0$, and a vector $q \in \mathbb{R}^N$, let $G : \mathbb{R}^d \to \mathbb{R}$ be defined as $G(t) = \sum_{i=1}^N q_i \cdot \mathsf{K}(s_i, t)$ denote the* kernel-density *of $t$ with respect*

---

**Algorithm 7** This algorithm is the query part of Theorem F.2.

1: **data structure** DYNAMICFGT
2: **procedure** KDE-QUERY($t \in \mathbb{R}^d$)
3:     Find the box $\mathsf{P}t \in \mathcal{C}_l$
4:     Compute                                            ▷ Takes $p^w$ time in total

$$G_p(t) \leftarrow \sum_{\beta \leq p} C_\beta(\mathcal{C}_l) \cdot \left( \frac{\mathsf{P}(t - t_{\mathcal{C}_l})}{\sqrt{\delta}} \right)^\beta$$

5:     **return** $G_p(t)$
6: **end procedure**
7: **procedure** QUERY($q \in \mathbb{R}^N$)
8:     INIT($\{s_j, j \in [N]\}, q$)                                     ▷ Takes $\widetilde{O}(N)$ time
9:     **for** $j \in [N]$ **do**
10:         $u_j \leftarrow$ LOCAL-QUERY($s_j$)                      ▷ $\|u - \mathsf{K}q\|_\infty \leq \varepsilon$
11:     **end for**
12:     **return** $u$
13: **end procedure**
14: **end data structure**

---

to $S$, where $\mathsf{K}(s_i, t) = f(\|s_i - t\|_2)$ for $f$ satisfying the properties in Definition 3.3 . There is a dynamic data structure that supports the following operations:

- INIT() *(Algorithm 6) Preprocess in* $N \cdot \log^{O(w)}(\|q\|_1/\varepsilon)$ *time.*

- KDE-QUERY($t \in \mathbb{R}^d$) *(Algorithm 7) Output* $\widetilde{G}$ *such that* $G(t) - \varepsilon \leq \widetilde{G} \leq G(t) + \varepsilon$ *in* $\log^{O(w)}(\|q\|_1/\varepsilon)$ *time.*

- INSERT($s \in \mathbb{R}^d, q_s \in \mathbb{R}$) *(Algorithm 8) For any source point* $s \in \mathbb{R}^d$ *and its charge* $q_s$, *update the data structure by adding this source point in* $\log^{O(w)}(\|q\|_1/\varepsilon)$ *time.*

- DELETE($s \in \mathbb{R}^d$) *(Algorithm 9) For any source point* $s \in \mathbb{R}^d$ *and its charge* $q_s$, *update the data structure by deleting this source point in* $\log^{O(w)}(\|q\|_1/\varepsilon)$ *time.*

- QUERY($q \in \mathbb{R}^N$) *(Algorithm 7) Output* $\widetilde{\mathsf{K}q} \in \mathbb{R}^N$ *such that* $\|\widetilde{\mathsf{K}q} - \mathsf{K}q\|_\infty \leq \varepsilon$, *where* $\mathsf{K} \in \mathbb{R}^{N \times N}$ *is defined by* $\mathsf{K}_{i,j} = \mathsf{K}(s_i, s_j)$ *in* $N \log^{O(w)}(\|q\|_1/\varepsilon)$ *time.*

## F.1 PROJECTION LEMMA

**Lemma F.3** (Hermite projection lemma in low-dimensional space, formal version of Lemma 3.5)**.** *Given a subspace* $B \in \mathbb{R}^{d \times w}$. *Let* $B^\top B = U\Lambda U^\top \in \mathbb{R}^{w \times w}$ *denote the spectral decomposition where* $U \in \mathbb{R}^{w \times w}$ *and a diagonal matrix* $\Lambda \in \mathbb{R}^{w \times w}$.

*We define* $\mathsf{P} = \Lambda^{-1/2} U^{-1} B^\top \in \mathbb{R}^{w \times d}$. *Then we have for any* $t, s \in \mathbb{R}^d$ *from subspace* $B$, *the following equation holds*

$$e^{-\|t-s\|_2^2/\delta} = \sum_{\alpha \geq 0} \frac{(\sqrt{1/\delta}\mathsf{P}(t-s))^\alpha}{\alpha!} h_\alpha(\sqrt{1/\delta}\mathsf{P}(t-s)).$$

*Proof.* First, we know that

$$\begin{aligned}
\mathsf{P}t &= \Lambda^{-1/2} U^{-1} B^\top t \\
&= \Lambda^{-1/2} U^{-1} B^\top B x \\
&= \Lambda^{-1/2} U^{-1} U\Lambda U^\top x \\
&= \Lambda^{-1/2} \Lambda U^\top x
\end{aligned}$$

---

**Algorithm 8** This algorithm is the update part of Theorem F.2.

---

1: **data structure** DYNAMICFGT
2: **members**                                   ▷ This is exact same as the members in Algorithm 6.
3:      $A_\alpha(\mathcal{B}_i), i \in [N(B)], \alpha \le p$
4:      $C_\beta(\mathcal{C}_i), i \in [N(C)], \beta \le p$
5:      $t_{\mathcal{C}_i}, i \in [N(C)]$
6:      $s_{\mathcal{B}_i}, i \in [N(B)]$
7: **end members**
8:
9: **procedure** INSERT($s \in \mathbb{R}^d, q \in \mathbb{R}$)
10:      Find the box $s \in \mathcal{B}$
11:      **for** $\alpha \le p$ **do**                                   ▷ we say $\alpha \le p$ if $\alpha_i \le p, \forall i \in [w]$
12:          Compute

$$A_\alpha^{\mathrm{new}}(\mathcal{B}) \leftarrow A_\alpha(\mathcal{B}) + \frac{(-1)^{\|\alpha\|_1} q}{\alpha!} (\frac{\mathsf{P}(s - s_\mathcal{B})}{\sqrt{\delta}})^\alpha$$

▷ Takes $p^w$ time

13:      **end for**
14:      Find $(2k+1)^w$ nearest target boxes to $\mathcal{B}$, denote by $\mathsf{nb}(\mathcal{B})$                ▷ $k \le \log(\|q\|_1/\varepsilon)$
15:      **for** $\mathcal{C}_l \in \mathsf{nb}(\mathcal{B})$ **do**
16:          **for** $\beta \le p$ **do**
17:              Compute

$$C_\beta^{\mathrm{new}}(\mathcal{C}_l) \leftarrow C_\beta(\mathcal{C}_l) + \frac{(-1)^{\|\beta\|_1}}{\beta!} \sum_{\alpha \le p} (A_\alpha^{\mathrm{new}}(\mathcal{B}) - A_\alpha(\mathcal{B})) \cdot H_{\alpha+\beta} \left( \frac{\mathsf{P}(s_\mathcal{B} - t_{\mathcal{C}_l})}{\sqrt{\delta}} \right)$$

▷ Takes $(2k+1)^w p^w$ time

18:          **end for**
19:      **end for**
20:      **for** $\alpha \le p$ **do**
21:          $A_\alpha(\mathcal{B}) \leftarrow A_\alpha^{\mathrm{new}}(\mathcal{B})$                                   ▷ Takes $p^w$ time
22:      **end for**
23:      **for** $\mathcal{C}_l \in \mathsf{nb}(\mathcal{B})$ **do**
24:          **for** $\beta \le p$ **do**
25:              $C_\beta(\mathcal{C}_l) \leftarrow C_\beta^{\mathrm{new}}(\mathcal{C}_l)$                          ▷ Takes $(2k+1)^w p^w$ time
26:          **end for**
27:      **end for**
28: **end procedure**
29: **end data structure**

---

$$= \Lambda^{1/2} U^\top x \tag{17}$$

where the first step follows from $\mathsf{P} = \Lambda^{-1/2} U^{-1} B^\top$, the second step follows from $t = Bx$ (since $t$ is from low dimension, then there is always a vector $x$), the third step follows $B^\top B = U \Lambda U^\top$, the forth step follows $U^{-1} U = I$, and the last step follows from $\Lambda^{-1/2} \Lambda = \Lambda^{1/2}$.

Compute the spectral decomposition $B^\top B = U \Lambda U^\top$, $U \in \mathbb{R}^{w \times w}$ is the orthonormal basis, $\Lambda = \mathrm{diag}(\lambda_1, \ldots, \lambda_k) \in \mathbb{R}^{w \times w}$. Let $u_i \in \mathbb{R}^w$ denote the vector that is the transpose of $i$-th row $U \in \mathbb{R}^{w \times w}$. Then we have

$$e^{-\|t-s\|_2^2/\delta} = e^{-(x-y)^\top B^\top B(x-y)/\delta}$$

$$= e^{-(x-y)^\top U \Lambda U^\top (x-y)/\delta}$$

$$= \prod_{i=1}^w \left( \sum_{n=1}^\infty \frac{1}{n!} (\sqrt{\lambda_i/\delta} \cdot u_i^\top (x-y))^n \cdot h_n(\sqrt{\lambda_i/\delta} \cdot u_i^\top (x-y)) \right)$$

$$= \sum_{\alpha \ge 0} \frac{\left( \sqrt{1/\delta} \Lambda^{1/2} U^\top (x-y) \right)^\alpha}{\alpha!} \cdot h_\alpha \left( \sqrt{1/\delta} \Lambda^{1/2} U^\top (x-y) \right)$$

---

**Algorithm 9** This algorithm is another update part of Theorem F.2.

---

1: **data structure** DYNAMICFGT
2: **members**
3:     $A_\alpha(\mathcal{B}_i), i \in [N(B)], \alpha \leq p$
4:     $C_\beta(\mathcal{C}_i), i \in [N(C)], \beta \leq p$
5:     $t_{\mathcal{C}_i}, i \in [N(C)]$
6:     $s_{\mathcal{B}_i}, i \in [N(B)]$
7:     $\delta \in \mathbb{R}$
8: **end members**
9:
10: **procedure** DELETE($s \in \mathbb{R}^d, q \in \mathbb{R}$)
11:     Find the box $s \in \mathcal{B}$
12:     **for** $\alpha \leq p$ **do**                          ▷ we say $\alpha \leq p$ if $\alpha_i \leq p, \forall i \in [w]$
13:         Compute

$$A_\alpha^{\text{new}}(\mathcal{B}) \leftarrow A_\alpha(\mathcal{B}) - \frac{(-1)^{\|\alpha\|_1} q}{\alpha!} \left( \frac{\mathsf{P}(s - s_\mathcal{B})}{\sqrt{\delta}} \right)^\alpha$$

▷ Takes $p^w$ time

14:     **end for**
15:     Find $(2k+1)^w$ nearest target boxes to $\mathcal{B}$, denote by $\mathsf{nb}(\mathcal{B})$          ▷ $k \leq \log(\|q\|_1/\varepsilon)$
16:     **for** $\mathcal{C}_l \in \mathsf{nb}(\mathcal{B})$ **do**
17:         **for** $\beta \leq p$ **do**
18:             Compute

$$C_\beta^{\text{new}}(\mathcal{C}_l) \leftarrow C_\beta(\mathcal{C}_l) + \frac{(-1)^{\|\beta\|_1}}{\beta!} \sum_{\alpha \leq p} (A_\alpha^{\text{new}}(\mathcal{B}) - A_\alpha(\mathcal{B})) \cdot H_{\alpha+\beta} \left( \frac{\mathsf{P}(s_\mathcal{B} - t_{\mathcal{C}_l})}{\sqrt{\delta}} \right)$$

▷ Takes $(2k+1)^w p^w$ time

19:         **end for**
20:     **end for**
21:     **for** $\alpha \leq p$ **do**
22:         $A_\alpha(\mathcal{B}) \leftarrow A_\alpha^{\text{new}}(\mathcal{B})$                          ▷ Takes $p^w$ time
23:     **end for**
24:     **for** $\mathcal{C}_l \in \mathsf{nb}(\mathcal{B})$ **do**
25:         **for** $\beta \leq p$ **do**
26:             $C_\beta(\mathcal{C}_l) \leftarrow C_\beta^{\text{new}}(\mathcal{C}_l)$                          ▷ Takes $(2k+1)^w p^w$ time
27:         **end for**
28:     **end for**
29: **end procedure**
30: **end data structure**

---

$$= \sum_{\alpha \geq 0} \frac{\left( \sqrt{1/\delta} \cdot \mathsf{P}(t - s) \right)^\alpha}{\alpha!} \cdot h_\alpha \left( \sqrt{1/\delta} \cdot \mathsf{P}(t - s) \right)$$

where the first step follows from changing the basis preserves the $\ell_2$-distance, the second step follows from $B^\top B = U \Lambda U^\top$, and the fifth step follows from Eq. (17).          □

### F.2    PROOF OF MAIN RESULT IN LOW-DIMENSIONAL CASE

*Proof of Theorem F.2.* **Correctness of KDE-QUERY.** Algorithm 6 accumulates all sources into truncated Hermite expansions and transforms all Hermite expansions into Taylor expansions via Lemma F.4. Thus it can approximate the function $G(t)$ by

$$G(t) = \sum_\mathcal{B} \sum_{j \in \mathcal{B}} q_j \cdot e^{-\|t - s_j\|_2^2/\delta}$$

$$= \sum_{\beta \leq p} C_\beta \left( \frac{\mathsf{P}(t - t_{\mathcal{C}})}{\sqrt{\delta}} \right)^\beta + \mathrm{Err}_T(p) + \mathrm{Err}_H(p)$$

where $|\mathrm{Err}_H(p)| + |\mathrm{Err}_T(p)| \leq Q \cdot \varepsilon$ by $p = \log(\|q\|_1 / \varepsilon)$,

$$C_\beta = \frac{(-1)^{\|\beta\|_1}}{\beta!} \sum_{\mathcal{B}} \sum_{\alpha \leq p} A_\alpha(\mathcal{B}) H_{\alpha+\beta} \left( \frac{\mathsf{P}(s_{\mathcal{B}} - t_{\mathcal{C}})}{\sqrt{\delta}} \right)$$

and the coefficients $A_\alpha(\mathcal{B})$ are defined as Line 24.

**Running time of KDE-QUERY.** In line 24, it takes $O(p^w N)$ time to compute all the Hermite expansions, i.e., to compute the coefficients $A_\alpha(\mathcal{B})$ for all $\alpha \leq p$ and all source boxes $\mathcal{B}$.

Making use of the large product in the definition of $H_{\alpha+\beta}$, we see that the time to compute the $p^w$ coefficients of $C_\beta$ is only $O(dp^{w+1})$ for each box $\mathcal{B}$ in the range. Thus, we know for each target box $\mathcal{C}$, the running time is $O((2k+1)^w dp^{w+1})$, thus the total time in line 30 is

$$O(N(C) \cdot (2k+1)^w dp^{w+1}).$$

Finally, we need to evaluate the appropriate Taylor series for each target $t_i$, which can be done in $O(p^w M)$ time in line 4. Putting it all together, Algorithm 6 takes time

$$O((2k+1)^w dp^{w+1} N(C)) + O(p^w N) + O(p^w M)$$
$$= O\left( (M+N) \log^{O(w)}(\|q\|_1 / \varepsilon) \right).$$

**Correctness of UPDATE.** Algorithm 8 and Algorithm 9 maintains $C_\beta$ as follows,

$$C_\beta = \frac{(-1)^{\|\beta\|_1}}{\beta!} \sum_{\mathcal{B}} \sum_{\alpha \leq p} A_\alpha(\mathcal{B}) H_{\alpha+\beta} \left( \frac{\mathsf{P}(s_{\mathcal{B}} - t_{\mathcal{C}})}{\sqrt{\delta}} \right)$$

where $A_\alpha(\mathcal{B})$ is given by

$$A_\alpha(\mathcal{B}) = \frac{(-1)^{\|\alpha\|_1}}{\alpha!} \sum_{j \in \mathcal{B}} q_j \cdot \left( \frac{\mathsf{P}(s_j - s_{\mathcal{B}})}{\sqrt{\delta}} \right)^\alpha.$$

Therefore, the correctness follows similarly from Algorithm 6.

**Running time of UPDATE.** In line 12, it takes $O(p^w)$ time to update all the Hermite expansions, i.e. to update the coefficients $A_\alpha(\mathcal{B})$ for all $\alpha \leq p$ and all sources boxes $\mathcal{B}$.

Making use of the large product in the definition of $H_{\alpha+\beta}$, we see that the time to compute the $p^w$ coefficients of $C_\beta$ is only $O(dp^{w+1})$ for each box $\mathcal{C}_l \in \mathsf{nb}(\mathcal{B})$. Thus, thus the total time in line 17 is

$$O((2k+1)^w dp^{w+1}).$$

**Correctness of QUERY.** To compute $\mathsf{K}q$ for a given $q \in \mathbb{R}^w$, notice that for any $i \in [N]$,

$$(\mathsf{K}q)_i = \sum_{j=1}^{N} q_j \cdot e^{-\|s_i - s_j\|_2^2 / \delta}$$
$$= G(s_i).$$

Hence, this problem reduces to $N$ KDE-QUERY() calls. And the additive error guarantee for $G(t)$ immediately gives the $\ell_\infty$-error guarantee for $\mathsf{K}q$.

**Running time of QUERY.** We first build the data structure with the charge vector $q$ given in the query, which takes $\widetilde{O}_d(N)$ time. Then, we perform $N$ KDE-Query, each takes $\widetilde{O}_d(1)$. Hence, the total running time is $\widetilde{O}_d(N)$.

We note that when the charge vector $q$ is slowly changing, i.e., $\Delta := \|q^{\mathrm{new}} - q\|_0 \leq o(N)$, we can UPDATE the source vectors whose charges are changed. Since each INSERT or DELETE takes $\widetilde{O}_d(1)$ time, it will take $\widetilde{O}_d(\Delta)$ time to update the data structure.

Then, consider computing $\mathsf{K}q^{\text{new}}$ in this setting. We note that each source box can only affect $\widetilde{O}_d(1)$ other target boxes, where the target vectors are just the source vectors in this setting. Hence, there are at most $\widetilde{O}_d(\Delta)$ boxes whose $C_\beta$ is changed. Let $\mathcal{S}$ denote the indices of source vectors in these boxes. Since

$$G(s_i) = \sum_{\beta \leq p} C_\beta(\mathcal{B}_k) \cdot \left( \frac{\mathsf{P}(s_i - s_{\mathcal{B}_k})}{\sqrt{\delta}} \right)^\beta,$$

we get that there are at most $\widetilde{O}_d(\Delta)$ coordinates of $\mathsf{K}q^{\text{new}}$ that are significantly changed from $\mathsf{K}q$, and we only need to re-compute $G(s_i)$ for $i \in \mathcal{S}$. If we assume that the source vectors are well-separated, i.e., $|\mathcal{S}| = O(\delta)$, the total computational cost is $\widetilde{O}_d(\Delta)$.

Therefore, when the change of the charge vector $q$ is sparse, $\mathsf{K}q$ can be computed in sublinear time. □

**Lemma F.4** (Truncated Hermite expansion with truncated Taylor expansion (low dimension version of Lemma E.5 )). *Let $G(t)$ be defined as Def E.1. For an integer $p$, let $G_p(t)$ denote the Hermite expansion of $G(t)$ truncated at $p$, i.e.,*

$$G_p(t) = \sum_{\alpha \leq p} A_\alpha H_\alpha \left( \frac{\mathsf{P}(t - s_{\mathcal{B}})}{\sqrt{\delta}} \right).$$

*The Taylor expansion of function $G_p(t)$ at an arbitrary point $t_0$ can be written as:*

$$G_p(t) = \sum_{\beta \geq 0} C_\beta \cdot \left( \frac{\mathsf{P}(t - t_0)}{\sqrt{\delta}} \right)^\beta,$$

*where the coefficients $C_\beta$ are defined as*

$$C_\beta = \frac{(-1)^{\|\beta\|_1}}{\beta!} \sum_{\alpha \leq p} (-1)^{\|\alpha\|_1} A_\alpha \cdot H_{\alpha+\beta} \left( \mathsf{P} \frac{(s_{\mathcal{B}} - t_{\mathcal{C}})}{\sqrt{\delta}} \right). \tag{18}$$

*Let $\mathrm{Err}_T(p)$ denote the error in truncating the Taylor series after $p^w$ terms, in the box $\mathcal{C}$ (that has center $t_{\mathcal{C}}$ and side length $r\sqrt{2\delta}$), i.e.,*

$$\mathrm{Err}_T(p) = \sum_{\beta \geq p} C_\beta \left( \frac{\mathsf{P}(t - t_{\mathcal{C}})}{\sqrt{\delta}} \right)^\beta.$$

*Then, we have*

$$|\mathrm{Err}_T(p)| \leq \frac{2 \sum_{j \in \mathcal{B}} |q_j|}{(1-r)^w} \sum_{l=0}^{w-1} \binom{w}{l} (1 - r^p)^l \left( \frac{r^p}{\sqrt{p!}} \right)^{w-l}$$

*where $r \leq 1/2$.*

*Proof.* We can write $C_\beta$ in the following way:

$$C_\beta = \frac{(-1)^{\|\beta\|_1}}{\beta!} \sum_{j \in \mathcal{B}} q_j \sum_{\alpha \leq p} \frac{(-1)^{\|\alpha\|_1}}{\alpha!} \left( \frac{\mathsf{P}(s_j - s_{\mathcal{B}})}{\sqrt{\delta}} \right)^\alpha \cdot H_{\alpha+\beta} \left( \frac{\mathsf{P}(s_{\mathcal{B}} - t_{\mathcal{C}})}{\sqrt{\delta}} \right)$$

$$= \frac{(-1)^{\|\beta\|_1}}{\beta!} \sum_{j \in \mathcal{B}} q_j \left( \sum_{\alpha \geq 0} - \sum_{\alpha > p} \right) \frac{(-1)^{\|\alpha\|_1}}{\alpha!} \left( \frac{\mathsf{P}(s_j - s_{\mathcal{B}})}{\sqrt{\delta}} \right)^\alpha \cdot H_{\alpha+\beta} \left( \frac{\mathsf{P}(s_{\mathcal{B}} - t_{\mathcal{C}})}{\sqrt{\delta}} \right)$$

$$= B_\beta - \frac{(-1)^{\|\beta\|_1}}{\beta!} \sum_{j \in \mathcal{B}} q_j \sum_{\alpha > p} \frac{(-1)^{\|\alpha\|_1}}{\alpha!} \left( \frac{\mathsf{P}(s_j - s_{\mathcal{B}})}{\sqrt{\delta}} \right)^\alpha \cdot H_{\alpha+\beta} \left( \frac{\mathsf{P}(s_{\mathcal{B}} - t_{\mathcal{C}})}{\sqrt{\delta}} \right)$$

$$= B_\beta + (C_\beta - B_\beta)$$

Next, we have

$$|\text{Err}_T(p)| \leq \left| \sum_{\beta \geq p} B_\beta \left( \frac{\mathsf{P}(t - t_\mathcal{C})}{\sqrt{\delta}} \right)^\beta \right| + \left| \sum_{\beta \geq p} (C_\beta - B_\beta) \cdot \left( \frac{\mathsf{P}(t - t_\mathcal{C})}{\sqrt{\delta}} \right)^\beta \right| \tag{19}$$

Using Lemma E.4, we can upper bound the first term in the Eq. (19) by,

$$\left| \sum_{\beta \geq p} B_\beta \left( \frac{\mathsf{P}(t - t_\mathcal{C})}{\sqrt{\delta}} \right)^\beta \right| \leq \frac{\sum_{j \in \mathcal{B}} |q_j|}{(1 - r)^w} \sum_{l=0}^{w-1} \binom{w}{l} (1 - r^p)^l \left( \frac{r^p}{\sqrt{p!}} \right)^{w-l} .$$

Since we can similarly bound $C_\beta - B_\beta$ as follows

$$|C_\beta - B_\beta| \leq \frac{1}{\beta!} K \cdot Q_B 2^{\|\beta\|_1/2} \sqrt{\beta!} \leq K Q_B \frac{2^{\|\beta\|_1/2}}{\sqrt{\beta!}},$$

we have the same bound for the second term

$$\left| \sum_{\beta \geq p} (C_\beta - B_\beta) \left( \frac{\mathsf{P}(t - t_\mathcal{C})}{\sqrt{\delta}} \right)^\beta \right| \leq \frac{\sum_{j \in \mathcal{B}} |q_j|}{(1 - r)^w} \sum_{l=0}^{w-1} \binom{w}{l} (1 - r^p)^l \left( \frac{r^p}{\sqrt{p!}} \right)^{w-l} .$$

$\square$

### F.3 DYNAMIC LOW-RANK FGT WITH INCREASING RANK

We further give an algorithm for FGT when the low-dimensional subspace is dynamic, i.e., the rank may increase with data insertions.

**Theorem F.5** (Low Rank Dynamic FGT Data Structure). *Let $W$ be a subspace of $\mathbb{R}^d$ with dimension $\dim(S) = w \ll d$. Given $N$ source points $s_1, \ldots, s_N \in W$ with charges $q_1, \ldots, q_N$, and $M$ target points $t_1, \ldots, t_M \in W$, a number $\delta > 0$, and a vector $q \in \mathbb{R}^N$, let $G : \mathbb{R}^d \to \mathbb{R}$ be defined as $G(t) = \sum_{i=1}^N q_i \cdot \mathsf{K}(s_i, t)$ denote the kernel-density of $t$ with respect to $S$, where $\mathsf{K}(s_i, t) = f(\|s_i - t\|_2)$ for $f$ satisfying the properties in Definition 3.3 . There is a dynamic data structure that supports the following operations:*

- INIT() *(Algorithm 6) Preprocess in $N \cdot \log^{O(w)}(\|q\|_1/\varepsilon)$ time.*

- KDE-QUERY$(t \in \mathbb{R}^d)$ *(Algorithm 7) Output $\widetilde{G}$ such that $G(t) - \varepsilon \leq \widetilde{G} \leq G(t) + \varepsilon$ in $\log^{O(w)}(\|q\|_1/\varepsilon)$ time.*

- INSERT$(s \in \mathbb{R}^d, q_s \in \mathbb{R})$ *(Algorithm 10) For any source point $s \in \mathbb{R}^d$ and its charge $q_s$, update the data structure by adding this source point in $\log^{O(w)}(\|q\|_1/\varepsilon)$ time. The subspace dimension $w$ may be increased by 1 if $s$ is not in the original subspace.*

- QUERY$(q \in \mathbb{R}^N)$ *(Algorithm 7) Output $\widetilde{\mathsf{K}q} \in \mathbb{R}^N$ such that $\|\widetilde{\mathsf{K}q} - \mathsf{K}q\|_\infty \leq \varepsilon$, where $\mathsf{K} \in \mathbb{R}^{N \times N}$ is defined by $\mathsf{K}_{i,j} = \mathsf{K}(s_i, s_j)$ in $N \log^{O(w)}(\|q\|_1/\varepsilon)$ time.*

*Proof.* Since Algorithm 10 updates $A_\alpha, C_\beta$ in the same way as Algorithm 8, the correctness of Procedures KDE-QUERY and QUERY follows similarly from Theorem B.5.

Furthermore, SCALE takes $O(wd + (N(B) + N(C)) \cdot p^w)$ time. For the correctness, we know that the rows of $\mathsf{P}$ form an orthonormal basis for the subspace. For a newly inserted point $s$, if it is not lie in the subspace, $(I - \mathsf{P})s$ gives a new basis direction. Therefore, we can easily update $\mathsf{P}$ by attaching this vector (after normalization) as a column. Then, we show that the intermediate variables $A_\alpha$ and $C_\beta$ can be correctly updated for the new subspace. For each source box $\mathcal{B}$ and each $w$-tuple $\alpha \leq p$, we have

$$A_{(\alpha,0)}^{\mathsf{new}}(\mathcal{B}) = \frac{(-1)^{\|\alpha\|_1} \cdot (-1)^i}{\alpha! \cdot i!} \sum_{j \in \mathcal{B}} q_j \cdot \left( \frac{x_j' - x_\mathcal{B}'}{\sqrt{\delta}} \right)^{(\alpha,i)} = A_\alpha(\mathcal{B}),$$

---

**Algorithm 10** This algorithm is the update part of Theorem F.5.

---

1: **data structure** DYNAMICFGT
2: **members**
3:     $k \in \mathbb{N}$ $\qquad\qquad\qquad\qquad\qquad\qquad\qquad\qquad\qquad$ ▷ Rank of $\text{span}(s_1, \ldots, s_N, t_1, \ldots, t_M)$
4:     $A_\alpha(\mathcal{B}_l), l \in [N(B)], \alpha \le p$
5:     $C_\beta(\mathcal{C}_l), l \in [N(C)], \beta \le p$
6:     $t_{\mathcal{C}_l}, l \in [N(C)]$
7:     $s_{\mathcal{B}_l}, l \in [N(B)]$
8:     $\mathsf{P} \in \mathbb{R}^{w \times d}$
9: **end members**
10:
11: **procedure** INSERT($s \in \mathbb{R}^d, q \in \mathbb{R}$)
12:     SCALE($s, q$)
13:     Find the box $s \in \mathcal{B}$
14:     **for** $\alpha \le p$ **do** $\qquad\qquad\qquad\qquad\qquad\qquad$ ▷ we say $\alpha \le p$ if $\alpha_i \le p, \forall i \in [w]$
15:         Compute

$$A_\alpha^{\text{new}}(\mathcal{B}) \leftarrow A_\alpha(\mathcal{B}) + \frac{(-1)^{\|\alpha\|_1} q}{\alpha!} \left(\frac{\mathsf{P}(s - s_\mathcal{B})}{\sqrt{\delta}}\right)^\alpha$$

$\qquad\qquad\qquad\qquad\qquad\qquad\qquad\qquad\qquad\qquad\qquad\qquad\qquad$ ▷ Takes $p^k$ time
16:     **end for**
17:     Find $(2k+1)^w$ nearest target boxes to $\mathcal{B}$, denote by $\mathsf{nb}(\mathcal{B})$ $\qquad$ ▷ $k \le \log(\|q\|_1/\varepsilon)$
18:     **for** $\mathcal{C}_l \in \mathsf{nb}(\mathcal{B})$ and $\beta \le p$ **do**
19:         Compute

$$C_\beta^{\text{new}}(\mathcal{C}_l) \leftarrow C_\beta(\mathcal{C}_l) + \frac{(-1)^{\|\beta\|_1}}{\beta!} \sum_{\alpha \le p} \left(A_\alpha^{\text{new}}(\mathcal{B}) - A_\alpha(\mathcal{B})\right) \cdot H_{\alpha + \beta}\left(\frac{\mathsf{P}(s_\mathcal{B} - t_{\mathcal{C}_l})}{\sqrt{\delta}}\right)$$

$\qquad\qquad\qquad\qquad\qquad\qquad\qquad\qquad\qquad\qquad\qquad\qquad$ ▷ Takes $(2k+1)^w p^w$ time
20:     **end for**
21:     **for** $\alpha \le p$ **do**
22:         $A_\alpha(\mathcal{B}) \leftarrow A_\alpha^{\text{new}}(\mathcal{B})$ $\qquad\qquad\qquad\qquad\qquad\qquad\qquad$ ▷ Takes $p^w$ time
23:     **end for**
24:     **for** $\mathcal{C}_l \in \mathsf{nb}(\mathcal{B})$ and $\beta \le p$ **do**
25:         $C_\beta(\mathcal{C}_l) \leftarrow C_\beta^{\text{new}}(\mathcal{C}_l)$ $\qquad\qquad\qquad\qquad\qquad\qquad$ ▷ Takes $(2k+1)^w p^w$ time
26:     **end for**
27: **end procedure**
28: **end data structure**

---

where $x'_j$ denotes the "lifted" point in the new subspace. And $A_{(\alpha,i)}^{\text{new}}(\mathcal{B}) = 0$ for all $i > 0$, since $(x'_j - x'_\mathcal{B})_{k+1} = 0$. Similarly, for each target box $\mathcal{C}$,

$$\begin{aligned}
C_{(\beta,i)}^{\text{new}}(\mathcal{C}) &= \frac{(-1)^{\|\beta\|_1}(-1)^i}{\beta! i!} \sum_\mathcal{B} \sum_{\alpha \le p} \sum_{j=0}^p A_{(\alpha,j)}^{\text{new}}(\mathcal{B}) H_{(\alpha+\beta, i+j)}\left(\frac{x'_\mathcal{B} - y'_\mathcal{C}}{\sqrt{\delta}}\right) \\
&= \frac{(-1)^{\|\beta\|_1}(-1)^i}{\beta! i!} \sum_\mathcal{B} \sum_{\alpha \le p} A_\alpha(\mathcal{B}) H_{\alpha+\beta}\left(\frac{x_\mathcal{B} - y_\mathcal{C}}{\sqrt{\delta}}\right) \cdot h_i(0) \\
&= \frac{(-1)^i}{i!} h_i(0) \cdot C_\beta(\mathcal{C}),
\end{aligned}$$

where the second step follows from $A_{(\alpha,i)}^{\text{new}}(\mathcal{B}) = A_\alpha(\mathcal{B}) \cdot \mathbf{1}_{i=0}$. Therefore, by enumerating all boxes $\mathcal{B}, \mathcal{C}$ and indices $\alpha, \beta \le p$, we can correctly compute $A_{(\alpha,0)}^{\text{new}}(\mathcal{B})$ and $C_{(\beta,i)}^{\text{new}}(\mathcal{C})$. Thus, we complete the proof of the correctness of Algorithm 11. $\qquad\qquad\qquad\qquad\qquad\qquad\qquad\qquad\qquad\qquad$ $\square$

**Algorithm 11** This algorithm is another part of Theorem F.5.

1: **data structure** DYNAMICFGT
2: **members**
3:     $w \in \mathbb{N}$                                                   ▷ Rank of $\mathrm{span}(s_1, \ldots, s_N, t_1, \ldots, t_M)$
4:     $A_\alpha(\mathcal{B}_l), l \in [N(B)], \alpha \le p$
5:     $C_\beta(\mathcal{C}_l), l \in [N(C)], \beta \le p$
6:     $t_{\mathcal{C}_l}, l \in [N(C)]$
7:     $s_{\mathcal{B}_l}, l \in [N(B)]$
8:     $\mathsf{P} \in \mathbb{R}^{w \times d}$
9: **end members**
10:
11: **procedure** SCALE($s \in \mathbb{R}^d, q \in \mathbb{R}$)
12:     **if** $s \in \mathrm{span}(P)$ **then**
13:         **pass**
14:     **else**
15:         $\mathsf{P} \leftarrow (\mathsf{P}, (I - \mathsf{P})s/\|(I - \mathsf{P})s\|_2), w \leftarrow w + 1$
16:         **for** $\mathcal{B}_l, l \in [N(B)]$ and $\mathcal{C}_l, l \in [N(C)]$ **do**
17:             $s_{\mathcal{B}_l} \leftarrow (s_{\mathcal{B}_l}, 0)$ and $t_{\mathcal{C}_l} \leftarrow (t_{\mathcal{C}_l}, 0)$
18:         **end for**
19:         Find the box $\mathcal{B}_{N(B)+1}$ of length $r\sqrt{\delta}$ containing $s$ and let $s_{\mathcal{B}_{N(B)+1}}$ be its center
20:         **for** $\alpha \le p \in \mathbb{N}^w$ and $\mathcal{B}_l, l \in [N(B)]$ **do**
21:             $A_{(\alpha,0)}(\mathcal{B}_l) \leftarrow A_\alpha(\mathcal{B}_l)$
22:         **end for**
23:         **for** $\beta \le p \in \mathbb{N}^w, 0 \le i \le p$ and $\mathcal{C}_l, l \in [N(C)]$ **do**
24:             $C_{(\beta,i)}(\mathcal{C}_l) \leftarrow \frac{(-1)^i}{i!} h_i(0) \cdot C_\beta(\mathcal{C}_l)$
25:         **end for**
26:     **end if**
27: **end procedure**
28: **end data structure**

## LLM USAGE DISCLOSURE

LLMs were used only to polish language, such as grammar and wording. These models did not contribute to idea creation or writing, and the authors take full responsibility for this paper's content.

