# OpenReview forum: "A Dynamic Low-Rank Fast Gaussian Transform"
_ICLR.cc/2026/Conference — Submitted to ICLR 2026_

### Official Review · Reviewer_GhF7 · 2025-10-26

**Soundness:** 3
**Presentation:** 3
**Contribution:** 2
**Rating:** 8
**Confidence:** 2

**Summary:**

This paper studies dynamic algorithms for high-precision Gaussian kernel density estimation/Fast Gaussian transform. In particular given a set of points $x_1,\ldots,x_n\in \mathbb{R}^{d}$ and associated charges $q_1,\ldots, q_n \in \mathbb{R}$, given a point $x\in \mathbb{R}^d$ the goal is to output an $\epsilon$ approximation to

 $\sum_{i=1}^n q_i e^{-\|x-x_i\|_2^2}$

 faster than linear in $n$ time. The famous fast multipole method achieves this in $log^{O(d)}(n/\epsilon))$ time but is not is a dynamic algorithm that supports updates to changing points $x_1,\ldots,x_n$.
The contribution of the paper is to present a dynamic algorithm for this problem under the assumption that the data points live in a $w$ dimensional space. In particular their algorithm supports two operations in $log^{O(w)}(n/\epsilon)$ time, the first is to accomodate updates to a point $x_i$ and associated charge $q_i$, and secondly output an $\epsilon$ approximation to

$\sum_{i=1}^n q_i e^{-\|x-x_i\|_2^2}$

for any $q$ in the same subspace. The main idea in their algorithm is a dynamic datastructure for maintaining the projected interaction rank between source and target boxes, decoupled into finite truncation of Taylor and Hermite expansion.

**Strengths:**

The main strength of the paper is to study dynamic algorithms for fast Gaussian transform that is an important algorithm for various fundamental tasks in machine learning and data science.

**Weaknesses:**

Perhaps a minor weakness of the paper is to study the problem under the assumption that the points and the query live in a low dimensional subspace. I further ask specific questions regarding this in the questions section.

**Questions:**

Regarding the weakness, it would be interesting to know more motivations behind why the assumption is a natural assumption. Moreover which parts of the proof breakdown when for eg. the query is not in the same subspace as the points. Furthermore are there any other natural assumptions under which dynamic algorithms could be developed ?

---

> ### Author Response · Authors · 2025-12-04
>
> We thank the reviewer for the positive assessment and the thoughtful questions. We appreciate the recognition of our contribution to dynamic algorithms for Fast Gaussian Transform.
>
> W1: The assumption that points and query live in a low dimensional subspace.
>
> The low-dimensional assumption is well-motivated by practical observations in machine learning. As noted by Cherapanamjeri & Nelson [1], kernel-based methods typically involve datasets where the intrinsic dimension $w$ is much smaller than the ambient dimension $d$. This is common in applications like online regression and federated learning where data often lies near a low-dimensional manifold. Our work directly addresses this gap between theory and practice by showing the computational cost can depend on $w$ instead of $d$.
>
> Q1: Why is the low-dimensional assumption natural, and which parts break down when query is not in the subspace?
>
> The assumption is natural because real-world high-dimensional data often has low intrinsic dimension due to underlying structure or correlations. When the query $t$ is not in the subspace $B$, Lemma 3.5 (Hermite projection lemma) breaks down. Specifically, the identity $e^{-\\|t-s\\|\_2^2/\\delta} = \\sum\_{\\alpha \\geq 0} (\\sqrt{1/\\delta}P(t-s))^\\alpha / \\alpha! \\cdot h\_\\alpha(\\sqrt{1/\\delta}P(t-s))$ requires both $t$ and $s$ to come from the subspace $B$ so that the projection $P$ preserves the $\\ell\_2$ distance $\\|t-s\\|\_2$. If $t$ is outside the subspace, the projection introduces error in the distance computation.
>
> Q2: Are there other natural assumptions for dynamic algorithms?
>
> This is an interesting direction. One possibility is assuming bounded aspect ratio or well-separated point sets, which could allow more efficient box decompositions. Another is assuming slowly-changing subspaces where rank increases are rare, as we partially address in Section 3.6. We leave a systematic study of alternative assumptions to future work.
>
> We hope this addresses the reviewer's concerns.
>
> ### Reference
> [1] Yeshwanth Cherapanamjeri and Jelani Nelson. "Uniform approximations for randomized Hadamard transforms with applications". STOC 2022.

---

### Official Review · Reviewer_rjDX · 2025-10-30

**Soundness:** 3
**Presentation:** 3
**Contribution:** 2
**Rating:** 2
**Confidence:** 4

**Summary:**

This manuscript outlines a scheme to make the FGT dynamic (in the sense that points are allowed to move). The method is also tailored to work with data that lies in a low-dimensional subspace and a scheme is sketched out that allows the subspace to (slowly) evolve.

**Strengths:**

The manuscripts contributions are technically sound (albeit quite dense). Moreover, the problem addressed (updating points) is of practical interest.

**Weaknesses:**

The main weakness of the manuscript is that its focus is on a highly practical algorithm (the FGT) and, yet, it contains no experiments or illustrations that highlight if the proposed scheme is actually practical. Absent such analysis, it is hard to advocate for the contribution of the manuscript. Even basic experiments would really help to strengthen the manuscript. Given the context of the manuscript I consider this a very significant weakness.

While purely theoretical papers can certainly be valuable, this work directly builds on the FGT and focuses the presentation as such---that makes it feel like an evolution of that algorithm that takes it from practical to (potentially) impractical. If there is a claim that the theoretical developments are of sufficient interest then there needs to be a stronger case made that they are broader or have particularly interesting technical developments (relative to prior work). The current presentation focuses so strongly on the FGT that it is hard to see this.

Lastly, while the specialization to low-dimensional data is good to have, it is not clear how interesting the result is in the exactly low-dimensional case. Perhaps I am missing something about the goals, but if the data is exactly low-dimensional then after an orthonormal basis $Q$ is computed for the subspace (which can be done "cheaply") isn't it just a matter of running all the FGT machinery (including the update scheme) on $x_i = Q^Ts_i$ since the FGT only depends on pairwise distances and $\lVert x_i - x_j\rVert_2 = \lVert s_i - s_j\rVert _2$? This is actually the scheme advocated for, but somehow lemma 3.5 and the surrounding text makes it seem more complicated than "invoke your existing FGT code on $x_i$" and that leads me to believe I may be missing something.

**Questions:**

Does the manuscripts perspective on low-dimensional data provide a path to situations where the data nearly lies on some low-dimensional subspace (i.e., not exactly) or some more general low-dimensional manifold?

---

> ### Author Response · Authors · 2025-12-04
>
> We thank the reviewer for the detailed feedback. We address each point below.
>
> W1: No experiments to show if the proposed scheme is practical.
>
> This is a theoretical paper, and experiments are out of our scope. We focus on the algorithmic and mathematical foundations of dynamic FGT, establishing rigorous complexity bounds for updates and queries. There is a long tradition of pure theory papers at top ML venues that do not include experiments [1,2,3]. Our contribution follows this tradition by providing the first sublinear-time dynamic algorithm for FGT, which is a foundational result. We believe the theoretical contribution stands on its own merit.
>
> W2: Need stronger case for theoretical contributions if no experiments.
>
> Our main theoretical contribution is non-trivial: we show how to maintain the Taylor-Hermite expansion coefficients $A\_\\alpha(B)$ and $C\_\\beta(C)$ under point insertions and deletions in $\\log\^{O(w)}(\\|q\\|\_1/\\varepsilon)$ time. The key challenge is that changing a single source point $s\_i \\in \\mathbb{R}\^d$ affects an entire row of the kernel matrix $K$, i.e., $n$ distances $\\|s\_i - s\_j\\|$. Our data structure overcomes this by exploiting the interaction rank structure and showing that only $O((2k+1)\^d)$ neighboring boxes need updates. This is fundamentally different from naive recomputation and cannot be achieved by standard dynamization techniques for decomposable problems (as we discuss in Section 1.1).
>
> W3: Low-dimensional case seems trivial since distances are preserved under orthonormal projection.
>
> The reviewer is correct that $\\|x\_i - x\_j\\|\_2 = \\|s\_i - s\_j\\|\_2$ when $x\_i = Q\^\\top s\_i$ for orthonormal $Q$. However, the difficulty arises in the dynamic setting when the subspace itself changes. When a new point $s$ arrives that is not in the current subspace, we must update $P$ and all intermediate variables $A\_\\alpha(B)$, $C\_\\beta(C)$. Section 3.6 and Appendix F.3 show how to lift existing variables to the new subspace efficiently. Specifically, we prove that $A\^{\\text{new}}\_{(\\alpha,0)}(B) = A\_\\alpha(B)$ and $C\^{\\text{new}}\_{(\\beta,i)}(C) = \\frac{(-1)\^i}{i!}h\_i(0) \\cdot C\_\\beta(C)$, which allows updates in $\\log\^{O(w)}(\\|q\\|\_1/\\varepsilon)$ time without recomputing everything. This is the main technical novelty of the low-rank extension.
>
> Q1: Does the perspective provide a path to nearly low-dimensional data or manifolds?
>
> This is an interesting direction. For approximately low-rank data, one could project onto the top $w$ principal components and treat the residual as noise. The error would scale with the spectral tail $\\sum\_{i>w} \\sigma\_i\^2$. For manifolds, local linear approximations could potentially be combined with our framework, though this requires careful handling of chart transitions. We leave these extensions as future work.
>
> Thanks again for the detailed review.
>
> ### Reference
>
> [1] Zeyuan Allen-Zhu, Yuanzhi Li, and Zhao Song. “On the convergence rate of training recurrent neural
> Networks.” NeurIPS 2019.
>
> [2] Zeyuan Allen-Zhu, Yuanzhi Li, and Zhao Song. "A convergence theory for deep learning via over-parameterization". ICML 2019.
>
> [3] Josh Alman and Zhao Song. “Fast attention requires bounded entries.” NeurIPS 2023.

---

### Official Review · Reviewer_Kcvf · 2025-10-31

**Soundness:** 3
**Presentation:** 3
**Contribution:** 2
**Rating:** 4
**Confidence:** 3

**Summary:**

The paper studies the problem of efficient matrix-vector multiplication in the case where the matrix is a kernel matrix (first in the Gaussian kernel case and then in the more general case of fast decay kernels) and the kernel points and the target vector may be dynamic over time. The authors builds on the fast Gaussian transform (FGT), which considers the static case where all points in the kernel matrix and the target vector are known in advance, and extend it towards a dynamic FGT algorithm. The algorithm leverages a novel rewriting of the original FGT approximation procedure, which reveals how certain structures can be pre-computed, thus lowering the cost of adding new points and producing an efficient procedure to adapt the data structure. The authors discuss several extensions towards the case of low-dimensional state/dynamic spaces.

**Strengths:**

* New efficient FGT algorithm for matrix-vector multiplication in the dynamic case for Gaussian kernels.
* The algorithm can add/remove source points and can perform kernel density estimation in logarithmic time exponential in the dimensionality of the subspace that all points below to.
* The algorithm can be adapted to a larger family of fast-decay kernels and to the general case of points belonging to a static or dynamic subspace.
* The derivation of the algorithm builds on a (at the best of my knowledge) novel treatment of the truncations of the kernel and Taylor expansions that allows for an efficient pre-computation of structures that remain constant through the include/delete operations.

**Weaknesses:**

* While the paper is theoretical in nature, it lacks of any numerical validation of the proposed method. Unfortunately, the theoretical analysis of these numerical methods often hide operations and implementation details that "in practice" tend to dominate performance in many real-world regimes. Furthermore, while the theory does not require a precise tuning of the parameters (eg, delta, k, r, eps) their choice can have a significant impact on the overall performance. While the authors provide some guidance in Remark 3.1, even a simple numerical validation would help in strengthening the current contribution, in particular given the focus on machine learning applications.
* In the literature review, the authors do not seem to discuss randomized linear algebra approaches to kernel learning. While this may be difficult to compare in practice (randomized vs deterministic), an extensive theoretical as well as practical literature is available following this approach and an extensive review would help. See e.g.
"Randomized Numerical Linear Algebra: Foundations & Algorithms" https://arxiv.org/abs/2002.01387
"Random Fourier Features for Kernel Ridge Regression: Approximation Bounds and Statistical Guarantees" https://arxiv.org/abs/1804.09893
"oASIS: Adaptive Column Sampling for Kernel Matrix Approximation" https://arxiv.org/pdf/1505.05208
"Fourier Sparse Leverage Scores and Approximate Kernel Learning" https://arxiv.org/abs/2006.07340
"Distributed Adaptive Sampling for Kernel Matrix Approximation" https://proceedings.mlr.press/v54/calandriello17a.html

**Questions:**

Please see weaknesses.

---

> ### Author Response · Authors · 2025-12-04
>
> We thank the reviewer for the careful reading and constructive feedback on our work.
>
> W1: While the paper is theoretical in nature, it lacks of any numerical validation of the proposed method.
>
> We focus on the algorithmic and mathematical foundations of dynamic FGT, establishing rigorous complexity bounds for updates and queries. There is a long tradition of pure theory papers at top ML venues that do not include experiments [1,2,3]. Our contribution follows this tradition by providing the first sublinear-time dynamic algorithm for FGT, which is a foundational result. We believe the theoretical contribution stands on its own merit.
>
> W2: In the literature review, the authors do not seem to discuss randomized linear algebra approaches to kernel learning.
>
> Thank you for pointing out these references. We will add a discussion of randomized linear algebra methods in the related work section. We note that our approach is fundamentally different from randomized methods like random Fourier features or Nystrom approximation. Our algorithm is deterministic, which is crucial for handling adaptive update sequences as noted in our paper (see the discussion after Theorem 1.1).
>
> We hope this addresses the reviewer's concerns.
>
> ### Reference
>
> [1] Zeyuan Allen-Zhu, Yuanzhi Li, and Zhao Song. “On the convergence rate of training recurrent neural
> Networks.” NeurIPS 2019.
>
> [2] Zeyuan Allen-Zhu, Yuanzhi Li, and Zhao Song. "A convergence theory for deep learning via over-parameterization". ICML 2019.
>
> [3] Josh Alman and Zhao Song. “Fast attention requires bounded entries.” NeurIPS 2023.

---

### Official Review · Reviewer_cz3u · 2025-10-31

**Soundness:** 3
**Presentation:** 3
**Contribution:** 3
**Rating:** 6
**Confidence:** 3

**Summary:**

This paper considers the well-known Fast Gaussian Transform technique for computing kernel density estimates, and for approximating matrix-vector multiplication with a kernel matrix. The paper describes a method to create a dynamic version of the algorithm with poly-logarithmic update and query times. As with the static Fast Gaussian Transform, the techniques require an exponential dependency on the dimension $s$. This paper additionally discusses the case where the data lies in a low-dimensional subspace and shows that the complexity scales with the intrinsic dimension of the data, rather than the ambient dimension.

**Strengths:**

Problems of matrix-vector multiplication with a kernel matrix, and of kernel density estimation are important and relevant to the machine learning community, and the Fast Gaussian Transform is an important technique. The contributions of this paper to create a dynamic version of the FGT are therefore relevant and significant. While the techniques are largely direct generalizations of the standard FGT, this is nonetheless an important problem and this paper is the first (to my knowledge) to consider the FGT in the dynamic setting. The paper is generally well presented.

**Weaknesses:**

Given that the FGT is fast in practice for low-dimensional data, the key weakness of this paper is the lack of empirical evaluation. A fast implementation of the described method would be of significant value.

Typos and minor points
- In the introduction, the notation changes unnecessarily from using $x_i$ to using $s_i$ for the source points.
- In the related works comparison with LSH-based KDEs, you should be aware of recent work which directly address the question of dynamizing these data structures [1, 2]. Note that these methods achieve sublinear update time, but not polylogarithmic.
- In the references, I could not find the paper 'Josh Alman, Gary Miller, Timothy Chu, Shyam Narayanan, Mark Sellke, and Zhao Song. Metric transforms and low rank representations of kernels. In arXiv preprint, 2021.'. What is the correct reference for Definition B.1?

[1] Laenen et al. "Dynamic Similarity Graph Construction with Kernel Density Estimation", ICML 2025.
[2] Liang et al. "Dynamic maintenance of kernel density estimation data structure: From practice to theory.", arXiv.

**Questions:**

- Does Theorem B.5 follow as a direct corollary of Theorem F.2, with $w = d$?
- What is the practical implication of this work? Can the algorithm described here be made to work fast in practice?

---

> ### Author Response · Authors · 2025-12-04
>
> We thank the reviewer for the positive feedback on the significance of our work and the recognition that this is the first paper to consider FGT in the dynamic setting.
>
> W1: Lack of empirical evaluation
>
> We focus on the algorithmic and mathematical foundations of dynamic FGT, establishing rigorous complexity bounds for updates and queries. There is a long tradition of pure theory papers at top ML venues that do not include experiments [1,2,3]. Our contribution follows this tradition by providing the first sublinear-time dynamic algorithm for FGT, which is a foundational result. We believe the theoretical contribution stands on its own merit.
>
> Q1: Does Theorem B.5 follow as a direct corollary of Theorem F.2 with $w = d$?
>
> Yes, Theorem B.5 is indeed a special case of Theorem F.2 when $w = d$. When the intrinsic dimension equals the ambient dimension, the projection matrix $P$ becomes the identity, and all the low-dimensional algorithms reduce to their full-dimensional counterparts. We state both theorems separately for clarity, as the full-dimensional case (Theorem B.5) is simpler and serves as a stepping stone to understand the low-rank result.
>
> Q2: What is the practical implication? Can the algorithm work fast in practice?
>
> The practical implication is that our algorithm enables efficient maintenance of kernel density estimation in streaming or online settings where data points are inserted or deleted over time. Applications include online regression, federated learning, and continual learning, where datasets evolve dynamically. While we do not provide experiments, the polylogarithmic update time $\log^{O(w)}(\|q\|\_1/\varepsilon)$ suggests that for low intrinsic dimension $w$, the algorithm should be efficient in practice. The static FGT is already widely used in practice, and our dynamic extension inherits its numerical properties while adding efficient update capabilities.
>
> We will add the references [4,5] on dynamizing LSH-based KDEs and, and fix the notation consistency and the missing reference in the revision. We thank the reviewer for pointing these out.
>
> ### Reference
>
> [1] Zeyuan Allen-Zhu, Yuanzhi Li, and Zhao Song. “On the convergence rate of training recurrent neural
> Networks.” NeurIPS 2019.
>
> [2] Zeyuan Allen-Zhu, Yuanzhi Li, and Zhao Song. "A convergence theory for deep learning via over-parameterization". ICML 2019.
>
> [3] Josh Alman and Zhao Song. “Fast attention requires bounded entries.” NeurIPS 2023.
>
> [4] Steinar Laenen, Peter Macgregor, He Sun. "Dynamic Similarity Graph Construction with Kernel Density Estimation", ICML 2025.
>
> [5] Jiehao Liang, Zhao Song, Zhaozhuo Xu, Junze Yin, Danyang Zhuo. "Dynamic maintenance of kernel density estimation data structure: From practice to theory." arXiv 2025.

---

### Official Review · Reviewer_mERe · 2025-10-31

**Soundness:** 3
**Presentation:** 3
**Contribution:** 3
**Rating:** 8
**Confidence:** 3

**Summary:**

The reviewed work provides a data structure and algorithm for "dynamicising" the fast Gauss transform. The FGT allow allows approximating kernel density estimators at a cost $\log(\epsilon)$ with the key insight being to Taylor approximate the function $x \mapsto \exp((x - s)^2 / (2 \sigma)$ at a set of output cell centers and precompute the sums of taylor coefficients for different source locations $s$.

The reviewed work allows for the dynamic addition of new evaluation locations x and source locations s by also taylor expanding the maps  $s \mapsto \exp((x - s)^2 / (2 \sigma)$ and summing coefficients over sources locations.

**Strengths:**

The proposed algorithm is natural and solves a conceivably important problem
The paper is well written: I had not studied the FGT before and the FGT was a valuable resource in understanding both the static FGT and the dynamic extension proposed by the authors.

**Weaknesses:**

Some numerical examples and code for an implementation would have been helpful to appreciate practicality of the proposed algorithm, although it does seem practical to me.

**Questions:**

Typos:

In line 123, "match the statistic FGT algorithm" should be "match the static FGT algorithm.

---

> ### Author Response · Authors · 2025-12-04
>
> We sincerely thank the reviewer for their careful reading of our manuscript and for the positive assessment of our work.
> We have corrected the typo in our revised paper.

---

### Meta-Review · Area_Chair_yWva · 2026-01-13

**Summary:**

This paper considers the problem of recomputing the Fast Gaussian Transform when the underlying data is changing. Under the assumption that the data lies on a k-dimensional subspace, the paper derives an efficient dynamic FGT algorithm and establishes its computational complexity. The primary weakness of the paper is lack of any numerical validation. The authors argue that "There is a long tradition of pure theory papers at top ML venues that do not include experiments" -- however this paper cannot be considered to be motivated by a pure theory problem, and even good theory papers do provide some evidence of correctness and insights via numerical experiments.

**Reviewer Concerns:**

See above

**Reviewer Scores:**

See above. The two negative reviews highlighting lack of experiments as a major weakness would not have changed their score.

---

### Decision · Program_Chairs · 2026-01-26

Reject